# From Many Voices to One: Statistically Principled Aggregation of LLM Judges

## Abstract

LLM-as-a-judge—often with multiple judges—is now the standard paradigm for scalable model evaluation. This strategy is known to suffer from biases, spurious correlations, confounding factors, etc., and many heuristic approaches have been proposed to tackle these. We address this problem from the point of view of proba­bilistic graphical models, enabling us to ***capture the challenges involved in using multiple judges in a principled way***. By considering Markov random fields (MRF) with multiple latent factors, we can model undesired correlations between judges, a latent unknown true notion of quality, and one or more additional latent distractors (for example, generation length). The key technical challenge is to identify and learn a higher-rank latent variable MRF, which we solve via a new approach that mixes sparse plus low-rank and tensor decompositions. This enables us to better understand the quality and behavior of judges, leading to improved evaluation capabilities. In addition, we show how to augment our approach via programmatic judges that can be cheaply constructed and added to standard model-based judges. Empirically, our framework, CARE (Confounder-Aware Aggregation for Reliable Evaluation), demonstrates consistent gains on diverse public benchmarks, reducing aggregation error by up to 25.15% and showing robust integration of programmatic judges. Additionally, CARE offers superior performance and efficiency compared to individual-judge intervention strategies. These results underscore CARE's ability to effectively model correlations and mitigate biases, leading to more accurate and robust aggregation of LLM judge scores.

## 1 Introduction

Large language models (LLMs) are the workhorse solution for automated evaluation of model generations. For example, using *LLM-as-a-judge* systems avoids incurring the cost and latency of expert annotation [1]. Given the ease of applying such tools, a common evaluation paradigm is to *ensemble multiple LLM judges* to form consensus evaluation scores [2]. While attractive, these approaches are unreliable. Judges can be individually inaccurate and suffer from biases, e.g., relying on spurious factors like position or verbosity [3, 4, 5]. Additionally, judge models are highly correlated (due to being trained on the same data), so that incorporating more judges may add no additional evaluation signal—or worse, boost confidence in an incorrect assessment [6, 7].

Many heuristic techniques have been proposed to mitigate these issues. Single judge bias-reduction methods include answer-order shuffling [8], prompt calibration [9, 10, 11], and fine-tuned evaluators (e.g., JudgeLM [12], PandaLM [5]). Ensembling methods aggregate judge scores via a simple majority vote or average [13] in the hope of reducing unreliability. Unfortunately, these approaches ***do not provide a general and principled way to improve LLM-as-a-judge frameworks***. Indeed, ad-hoc approaches target one spurious factor (e.g., generation length [3]) and leave others in place, or

make implicit assumptions that are unlikely to hold (e.g., majority vote and unweighted averages assume access to independent and equally reliable judges).

These difficulties motivate the need for a *general* and *principled* approach to LLM-as-a-judge ensembles. We provide one through the lens of probabilistic graphical models—a classic framework that can be used for modeling and aggregating viewpoints. Concretely, we recast multi-judge evaluation as probabilistic inference in a ***higher-rank latent variable Markov Random Field (MRF)***. This enables us to model and deal with key challenges in LLM-as-a-judge ensembles:

- **No access to ground-truth scores**: One latent variable in the MRF represents a ground-truth quality for the generation being evaluated; we have *no access to it* and never observe it.
- **Unknown spurious factors**: Other latent MRF components model unknown and general distractors or spurious correlations that are associated with—but not causal—to generation quality. These might include generation length, verbosity, and other factors.
- **Complex correlations**: Judges may have correlations beyond their voting behavior, due to the use of shared data for training or shared base models. These correlations are flexibly modeled by MRF interactions between variables corresponding to judges.

Higher-rank latent variable MRFs provide a ***principled and general recipe to automated model-based evaluation***. The recipe is to learn the MRF (i.e., learn its parameters, including those for the latent variables, from observed data—LLM votes) then compute a posterior estimate of the latent ground-truth quality. However, learning such higher-rank latent MRFs is challenging. We must address **1)** how can we learn the model parameters despite never observing any latent variable, and **2)**: how can we identify which latent corresponds to a ground-truth quality score (rather than spurious factors)?

We tackle this technical challenge with a two-pronged approach. First, to address 1), we introduce a novel two-stage technique to learn higher-rank latent MRFs. It ***combines a sparse plus low-rank decomposition that partially recovers the model with a second tensor decomposition step to fix the remaining parameters***. While each approach has been individually used to learn latent factor models in more limited settings, our new combined approach is substantially more general. Second, to handle 2), we introduce a variety of approaches that boost identifiability, enabling us to distinguish between latent variables corresponding to ground-truth scores versus spurious factors or confounders.

In addition to our basic estimator, we develop an adaptive approach that *augments an existing set of judges with new, generated judges*. The augmented evaluators we focus on in particular are *programmatic judges*—programs that can perform evaluation that are themselves the output of LLMs. We find that such programmatic judges enable (1) boosting the signal for evaluation and (2) facilitate the expansion of the judge set, leading to improved accuracy and robustness.

**Summary of Contributions.**

1. We propose CARE, the first *confounder-aware aggregation* framework that explicitly models shared latent confounders among LLM judges, unifying single-judge debiasing with principled statistical fusion.
2. We prove identifiability and derive finite-sample error bounds, showing that our estimator can reliably aggregate judge scores even when confounders are non-trivial.
3. We characterize the inherent model misspecification error incurred by methods ignoring confounders, demonstrating CARE's advantage over independence-based competitors.
4. We demonstrate consistent gains on diverse public benchmarks, reducing aggregation error by up to **25.15%** and proving *more performant and efficient* than individual-judge intervention strategies.
5. We show that CARE *robustly integrates* programmatic judges and supports *progressive expansion* of the evaluator pool, *consistently outperforming baseline aggregation methods*.

By explicitly modeling confounders during aggregation, our framework offers a principled alternative to current heuristic pipelines and substantially enhances the reliability of LLM-as-a-judge.

## 2   Background and Overview

We start with brief background on automated evaluation and probabilistic graphical models.

**LLM-as-a-judge.** The goal of these techniques is to efficiently and cheaply evaluate model generations. Large language models can act as inexpensive, fast proxies for human raters by returning (i) *scalar quality scores* (e.g., 1–10 Likert or percentile ranks) [12, 5, 4], (ii) *pairwise preferences*

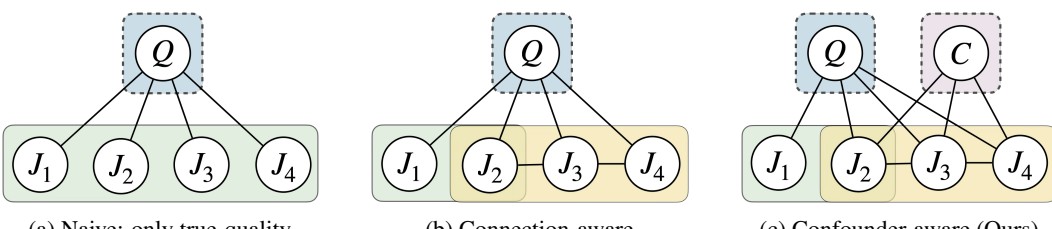

(a) Naive: only true-quality     (b) Connection-aware     (c) Confounder-aware (Ours)

Figure 1: **Graphical models for aggregating judge scores under different structural assumptions.** **(a)** A naive model assumes scores reflect only a true latent quality ($Q$) and that all judges are equally reliable and represent independent views. **(b)** Connection-aware approach models intra-judge interactions ($J_2 - J_3 - J_4$), but still assumes the presence of a single latent quality score. **(c)** Our Confounder-aware model introduces additional latent confounders ($C$) influencing judge scores.

that indicate which of two candidate answers is better—an output format popularized by RLHF pipelines [14, 15], and (iii) *categorical labels* such as error type, topic tag, or correctness flags [16, 8]. As individual LLM judges are often biased, recent work [17] deploys *multiple* LLM judges and aggregates their opinions—via majority vote, average pooling, or other techniques—to boost robustness and accuracy. Our framework builds on this line of work *but seeks a more principled approach to multi-judge aggregation that explicitly models shared confounders and correlated errors.*

**Graphical Models and Latent-Variable MRFs.** Graphical models represent conditional independence in multivariate distributions, with Markov Random Fields (MRFs) being particularly valuable due to their effective structure learning and efficient inference capabilities, enabling the discovery of meaningful dependency structures from data for probabilistic reasoning at scale. In our LLM-as-a-judge setting, we employ MRFs to jointly model judge scores ($J$), confounding factors ($C$), and latent quality variables, allowing us to capture intricate dependencies among LLM evaluations while maintaining efficient inference and learnability. When key influences are unobserved, such as the true quality signal, augmenting an MRF with latent nodes allows for the recovery of this hidden structure or "ground-truth" variables from noisy observations. This latent-variable MRF perspective is crucial in our context, offering a principled method to estimate the latent, true-quality signal from observable judges' scores while accounting for correlated judging errors.

## 3 CARE: Confounder-Aware Aggregation for Reliable Evaluation

We introduce CARE (Confounder-Aware Aggregation for Reliable Evaluation), our graphical model-based aggregation framework that robustly estimates the true quality of LLM-as-a-judge assessments. Our framework explicitly models the influence of a latent true-quality variable and additional latent confounders on the observed scores provided by multiple judges.

### 3.1 Graphical Model Framework And Assumptions

For each prompt-response pair, we observe scores $J = (J_1, \ldots, J_p)^\top$ from $p$ judges. We assume these observed scores depend on latent variables including one *true quality variable* $Q$ and one or more *confounders* $C = (C_1, \ldots, C_k)$, which we define as $H = (Q, C)$. Our graphical model encodes the conditional independence structure among the nodes in $(J, Q, C)$: if there is no edge between a pair of nodes, they are independent conditioned on the other nodes. An example is shown on the right in Fig. 1. We assume this structure is *sparse*; i.e., there are not too many edges in the graph, and make this precise later on.

This framework is quite general and is compatible with a variety of distributions. For example, we may take $J, Q, C$ to involve discrete variables, Gaussians, or mixed models. We can take the model to be an MRF or alternatively a mixture model. Our approaches are compatible with a broad range of choices, with practitioners able to select the most suitable modeling assumptions for their settings.

**Goals and Assumptions.** Under the chosen modeling assumptions, our goal is to learn the distribution over $J, Q, C$. This involves handling ***three challenges***. First, **C1**: *we never observe the latents in H*—neither ground truth nor confounders. Second, **C2**: *we cannot assume any particular interaction* in the graph. Third, **C3**: even if we recover the model parameters, we must be able to distinguish

---

**Algorithm 1** CARE: Confounder-Aware Aggregation for Reliable Evaluation

---

**Input:** Score matrix $J \in \mathbb{R}^{n \times p}$, parameters $(\gamma_n, \tau)$, decomposition method $\mathcal{D} \in \{\text{SVD}, \text{Tensor}\}$
**Output:** Estimated True Quality $\{\hat{q}^{(i)}\}_{i=1}^n$

1: **Graph Sparse Structure Estimation:** Compute appropriate observed matrix $f(J)$.
2: **Sparse + low-rank decomposition:**

$$(\hat{S}, \hat{L}) \leftarrow \underset{S,L}{\arg\min} \; \tfrac{1}{2}\|f(J) - S - L\|_F^2 + \gamma_n(\|S\|_1 + \tau\|L\|_*)$$

3: **Latent Factor Extraction:**
4: **if** $\mathcal{D} = \text{SVD}$ **then**                                            ▷ Fully Gaussian scenario
5:     Compute $U \Lambda U^\top \leftarrow \text{SVD}(\hat{L})$, where $U \in \mathbb{R}^{p \times h}$
6: **else if** $\mathcal{D} = \text{Tensor}$ **then**                                            ▷ Binary-Gaussian mixture scenario
7:     Partition judges into independent groups using $\hat{S}$
8:     Form empirical third-order tensor from judge groups
9:     Run tensor decomposition, obtain latent conditional means $\mu_{qc}$ and mixture proportions $\pi_{qc}$
10: **end if**
11: **Symmetry Breaking:** Identify the true-quality factor using heuristics described in §3.3
12: **Latent Quality Estimation:** Use the identified quality factor, compute $\hat{q}^{(i)}$ for each example, where $\hat{q}^{(i)} = P(Q = 1 \mid J_i)$ for mixture model or $\hat{q}^{(i)} = \mathbb{E}[Q \mid J]$ for fully gaussian

---

127  between $Q$ and the confounders $C$ *to identify the model*. The latter is required to discover **which**
128  **latent is the ground-truth quality—and which is a confounder**. Once these obstacles are overcome,
129  we seek to perform aggregation, e.g., compute a posterior $P(Q|J)$, the Bayesian estimate for the
130  latent true quality conditioned on all observable judge scores.

131  In the following, we will work under the assumption that the judge scores $J$ conditioned on the latents
132  form a multivariate Gaussian distribution, i.e., $J \mid H \sim \mathcal{N}(\mu_H, \Sigma)$, where $\mu_H$ is the conditional
133  mean of observable variables. We defer other scenarios to the Appendix.

## 3.2  CARE Algorithm

135  The idea behind CARE is to examine two techniques, each of which is stymied by one of the
136  obstacles **C2** or **C3** and to *delicately combine them in a novel way*. First, the sparsity of the
137  conditional independence graph is encoded into an two-dimensional object that can be empirically
138  estimated (e.g., the observable covariance matrix, or a cross-moment matrix). However, the presence
139  of the latent variables (**C1**) obscures this structure—but a *sparse + low-rank decomposition* can
140  reveal it [18]. However, while we can decompose the resulting low-rank term via SVD in the hope of
141  identifying the model, we can only do so *up to rotations*. Therefore we are blocked by **C3**.

142  Conversely, tensor product decompositions [19] exploit tensor rigidity to enable this decomposition
143  to be uniquely identified. However, for these techniques the judges must be independent conditioned
144  on the latents—and we cannot assume this by **C2**.

145  CARE (Algorithm 1) combines these approaches. First, it estimates the underlying graph structure
146  from the observed judge scores via the sparse + low-rank decomposition, overcoming **C1** and **C2**. It
147  then uses recovered sparse term to estimate the graph and discover subsets of judges with sufficient
148  conditional independence. These sets are then used to construct a tensor that can be decomposed via
149  standard approaches (e.g., tensor power method) to recover the model, mitigating **C3**.

150  This procedure is then followed by a symmetry-breaking step. This requires a weak assumption on
151  the quality of the judges; in practice, even this assumption can be removed by employing simple
152  heuristics to identify the true-quality factor among the latent factors. Finally, we aggregate judge
153  scores into robust evaluations by weighting according to loadings from the identified quality factor.

154  We study two special cases to build our intuition; more general settings are shown in the Appendix.

**CARE For Gaussian Mixtures.** We have binary latents $(Q, C)$ with $\Pr\big(Q = q,\, C = c\big) = \pi_{qc}$, where the judges follow a Gaussian conditional distribution with mean $\mu_{qc} \in \mathbb{R}^p$ and covariance $\Sigma$:

$$J \mid (Q = q,\, C = c) \sim \mathcal{N}\big(\mu_{qc}, \Sigma\big), \qquad (q, c) \in \{0, 1\}^2.$$

Here, performing the sparse + low-rank decomposition and obtaining $\hat{L}$ is insufficient: the eigendecomposition of $\hat{L}$ does not directly yield identifiable latent-judge connections. We rely on third-order tensor statistics to identify conditional distributions explicitly:

$$\mathbb{E}(X_1 \otimes X_2 \otimes X_3 \mid Q, C) = \mathbb{E}(X_1 \mid Q, C) \otimes \mathbb{E}(X_2 \mid Q, C) \otimes \mathbb{E}(X_3 \mid Q, C),$$

where judges are partitioned into independent groups $X_1, X_2, X_3$ using the learned sparse structure $\hat{S}$. Performing a tensor decomposition yields the conditional means $\mu_{qc}$ and mixture proportions $\pi_{qc}$. Then, applying Bayes' rule allows estimation of latent variables given observed scores:

$$P(Q = 1|J) \propto \pi_{10}\mu_{10} + \pi_{11}\mu_{11}. \tag{1}$$

**CARE for Fully Gaussian Models.** Under the fully Gaussian assumption, latent variables $H$ are continuous, and the inverse covariance matrix (the *precision* matrix) encodes independence:

$$\Sigma = \mathrm{Cov}\big[(J, H)^\top\big], \quad \Sigma^{-1} = K = \begin{pmatrix} K_{JJ} & K_{JH} \\ K_{HJ} & K_{HH} \end{pmatrix}, \quad S = K_{JJ}, \quad L = K_{JH} K_{HH}^{-1} K_{HJ}.$$

If assuming connections $K_{JH}$ between latent variables and judges are orthogonal and no direct connections among latent variables (i.e. $K_{HH}$ is diagonal), the low-rank matrix $\hat{L}$ admits eigendecomposition $\hat{L} = U \Lambda U^\top$, where eigenvectors in $U$ directly correspond to latent-judge edges ($K_{JH}$), and eigenvalues correspond to $K_{HH}$. Each eigenvector represents how one latent variable influences observable judges. With these edges recovered, the conditional mean of true quality $Q$ can be estimated by $\mathbb{E}(Q \mid J) = K_{QQ}^{-1} K_{QJ} J$, a weighted linear combination of observed scores.

The fully Gaussian model prevents decomposing the low-rank term uniquely (due to rotational invariance). This holds regardless of whether we apply SVD or a tensor decomposition, leading to the special handling in Algorithm 1. As a result, in this case, orthogonal and independent latent assumptions are needed for identifying the latent-judge connection. This works the best when each judge is connected to exactly one latent variable. If a judge depends on *both* the confounder $C$ and the true quality $Q$ with comparable weights, the recovered columns $\{\hat{\mu}_r\}$ are only identifiable up to an arbitrary rotation, causing estimation errors.

### 3.3 Heuristics for Identifiability and Robust Estimation

Any instantiation of CARE will require symmetry-breaking procedures for latent variable identifiability. For example, the fully Gaussian case needs a heuristic to identify the true-quality direction among latent factors, distinguishing $Q$ from confounders $C$. In the binary-Gaussian mixture scenario, an additional step resolves ambiguity between latent states ($Q = 0$ vs. $Q = 1$). Doing so will require additional information that can come from modeling assumptions, the use of ground-truth samples, or heuristics. We detail some examples below:

**Identifying True-Quality Factor for Joint-Gaussian Model.** We introduce heuristics particularly aimed at distinguishing the true-quality latent variable from confounding latent variables. First, the *human-anchor criterion* leverages a small validation set containing human ratings. By including these human judgments in the graphical model, we anchor the latent quality variable to ground truth by selecting the latent factor exhibiting the strongest connection to the human evaluations. Second, we apply a *loading balance heuristic*, identifying the true-quality factor as one that loads broadly and with similar magnitude across all competent judges. Conversely, factors dominated by a few judges typically indicate shared confounding rather than true quality.

**Identifying Latent States for Mixed Model.** In scenarios such as the tensor-based method, symmetry breaking additionally involves distinguishing latent states corresponding to different quality levels (e.g., $Q = 0$ versus $Q = 1$). In practice, we can use known labeled samples (such as high-quality examples) to anchor and identify latent-state configurations. By comparing different latent configurations with these known labeled samples, we select the latent-state assignment that best aligns with empirical observations, effectively removing latent state ambiguity.

## 4    Theoretical Analysis

We provide the following theoretical guarantees for our Algorithm 1.

**Identifiability of the Latent Structure.** To ensure identifiability of the latent structure, we introduce assumptions on latent independence and orthogonality of latent-observable connections. Under these assumptions, we prove exact recovery of the latent directions, as well as stability under mild perturbations from orthogonality (see Appendix D.2).

**Sample Complexity Bound.** We derive sample complexity bounds for consistent estimation of latent-observable connections, demonstrating how estimation accuracy depends on factors like eigengaps and manifold curvature (Appendix D.3).

**Model Misspecification Error.** We analyze errors arising from model misspecification—specifically, the bias introduced when confounding latent factors are omitted—and provide explicit bounds on the resulting errors in estimated conditional means (Appendix D.4).

## 5    Experimental Results

We evaluate the effectiveness of CARE across diverse experimental setups, encompassing synthetic, semi-synthetic, and real-world scenarios. Our goal is to validate the following key claims:

- **Improving aggregation of LLM judge:** CARE produces more accurate and robust aggregate scores from multiple LLM judges compared to existing methods. (Section 5.1)

- **Effective Integration of Program Judges:** CARE integrates programmatic judges, known to have high bias, by explicitly modeling their biases [20] (Section 5.2).

- **Evolving Jury via Progressive Program Judge Expansion:** CARE effectively incorporates an expanding pool of judges, demonstrating consistent improvements in aggregation performance as judges are progressively added (Section 5.3).

- **Greater Robustness than Individual Intervention:** CARE is competitive against interventions at the individual judge level, which typically require extensive manual tuning (Section 5.4).

- **Demonstrating Robustness under Controlled Confounding Factors:** CARE remains accurate when evaluations are deliberately affected by controlled biases, as demonstrated by the semi-synthetic data from [8] (Section 5.5).

- **Validating Theoretical Results in a Fully Controlled Setting:** We empirically validate our theoretical results through synthetic experiments (Section 5.6).

**Datasets & Metrics.** We use FeedbackQA [21], UltraFeedback [22], and HelpSteer2 [23] datasets for response scoring. Performance is benchmarked using Mean Absolute Error (MAE) to measure numerical accuracy and Kendall's $\tau$ rank correlation [24] to evaluate ranking consistency, accommodating variations in judge scales and calibration.

**Baselines.** We compare CARE to following baseline aggregation methods: (i) majority voting (MV), (ii) simple averaging (AVG) [13], (iii) discrete-based weak supervision (WS) [25], and (iv) continuous-based weak supervision (UWS) [26].

**LLM Judges.** We consider the following LLMs as judges to score responses: `Llama-3.2-1B` [27], `Llama-3.1-8B-Instruct` [27], `Mistral-7B-Instruct-v0.3` [28], `Qwen3-0.6B` [29], `Qwen3-1.7B` [29], `Qwen3-4B` [29], `Qwen3-8B` [29], `Phi-4-mini-instruct` [30], `gemma-3-1b-it` [31], `gemma-3-4b-it` [31].

### 5.1    Improving Aggregation of LLM judges

**Setup.** We compare aggregation methods using the 10 LLM judges listed above. To ensure consistency, we adapt the prompt template from [32], modifying it to fit our experimental setup. The exact used prompt is provided in Appendix E.

**Results.** We present aggregation performance in Table 1. The CARE approach consistently outperforms baseline methods. Specifically, CARE achieves the lowest MAE on FeedbackQA (0.7866) and UltraFeedback (0.6379), outperforming the majority vote (MV) baseline by **10.74%** and **25.15%**,

Table 1: Aggregation performance across different datasets, measured by MAE and Kendall's $\tau$ CARE outperforms baseline methods in most cases.

| | FeedbackQA | | HelpSteer2 | | UltraFeedback | |
|---|---|---|---|---|---|---|
| | MAE ($\downarrow$) | $\tau$ ($\uparrow$) | MAE ($\downarrow$) | $\tau$ ($\uparrow$) | MAE ($\downarrow$) | $\tau$ ($\uparrow$) |
| MV | 0.8812 | 0.3703 | 0.9951 | 0.1629 | 0.8522 | 0.2985 |
| AVG | 0.8492 | 0.4497 | 0.9822 | 0.1611 | 0.6860 | 0.3621 |
| WS | 0.8144 | 0.4401 | 1.3030 | 0.1511 | 1.1603 | 0.3306 |
| UWS | 0.9051 | **0.4580** | 0.9849 | 0.1697 | 0.6794 | 0.3669 |
| CARE | **0.7866** | 0.4542 | **0.9742** | **0.1805** | **0.6379** | **0.3806** |

Table 2: Performance on different datasets using both LLM and program judges. Program judges are beneficial in FeedbackQA but may introduce noise in HelpSteer2 and UltraFeedback. In both cases, CARE consistently outperforms other baselines.

| | FeedbackQA | | HelpSteer2 | | UltraFeedback | |
|---|---|---|---|---|---|---|
| | MAE ($\downarrow$) | $\tau$ ($\uparrow$) | MAE ($\downarrow$) | $\tau$ ($\uparrow$) | MAE ($\downarrow$) | $\tau$ ($\uparrow$) |
| MV | 0.8607 | 0.3815 | 1.0244 | **0.1465** | 0.8751 | 0.3179 |
| AVG | 0.8128 | 0.4671 | 1.1012 | 0.1268 | 1.0371 | **0.3733** |
| UWS | 0.8179 | **0.4816** | 0.9992 | 0.1040 | 0.9534 | 0.3047 |
| CARE | **0.7582** | 0.4796 | **0.9800** | 0.1398 | **0.7351** | 0.3520 |

respectively. These gains highlight CARE's ability to model correlations among LLM judges and mitigate compounding biases.

## 5.2 Effective Integration of Program Judges

**Setup.** We integrate our LLM-based evaluators with ten program judges, each encoding their evaluation logic in program code and synthesized by OpenAI's GPT-4o [33]. These judges are designed to assess response quality through specific, individual criteria, such as *structure, readability, safety, relevance, and factuality*. While cost-effective to construct them, their deterministic nature may introduce systematic biases, potentially leading to noisy signals. Details of program judge generation process are provided in Appendix E.

**Results.** Table 2 presents the integration results. Adding program judges enhance performance on FeedbackQA, where CARE achieves the lowest MAE (0.7582) and highest $\tau$ (0.4796), outperforming the MV baseline's MAE by **11.92%**. However, performance declines on Help-Steer2 and UltraFeedback, where CARE records MAEs of 0.9800 and 0.7351, respectively, still outperforming MV by **4.33%** and **15.99%**. Despite these variations, CARE consistently exceeds baselines on MAE across all datasets, demonstrating its effectiveness when encountering noisier signals for aggregation.

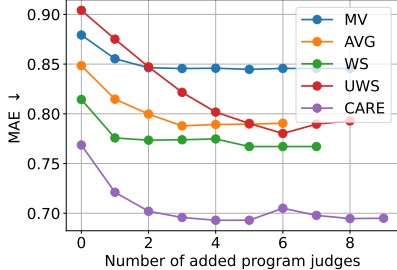

Figure 2: Progressive judge selection on the FeedbackQA dataset. CARE robustly integrates new judges and consistently outperforms baseline aggregation methods.

## 5.3 Progressive Judge Expansion

**Setup.** Next, we start with a fixed set of LLM judges and progressively add program judges from a pool of 23. At each step, we greedily select the program judge that yields the largest improvement in the validation of MAE. The process stops when no further reduction in validation MAE is observed. We evaluate aggregation methods as in previous sections, using FeedbackQA, where program judges were most beneficial.

**Results.** Figure 2 shows the experimental result. CARE achieves consistently lower error as more program judges are added, highlighting its ability to adaptively improve with additional supervision. This points to a promising direction for developing dynamic, expandable judge ensembles.

Table 3: Comparison with aggregation methods using individually intervened LLM judges. While other baselines aggregate scores from debiased LLM judges, CARE operates directly on raw outputs.

| | FeedbackQA | | HelpSteer2 | | UltraFeedback | |
|---|---|---|---|---|---|---|
| | MAE ($\downarrow$) | $\tau$ ($\uparrow$) | MAE ($\downarrow$) | $\tau$ ($\uparrow$) | MAE ($\downarrow$) | $\tau$ ($\uparrow$) |
| MV | 0.8004 | 0.9640 | 0.9951 | 0.1629 | 0.8562 | 0.2799 |
| AVG | 0.8029 | 0.4412 | 0.9822 | 0.1611 | 0.6801 | 0.3704 |
| WS | **0.7674** | 0.4429 | 1.3030 | 0.1511 | 1.1516 | 0.3588 |
| UWS | 0.8117 | 0.4390 | 0.9849 | 0.1697 | 0.6683 | 0.3782 |
| CARE | 0.7866 | **0.4542** | **0.9742** | **0.1805** | **0.6379** | **0.3806** |

Table 4: Robustness to artificially injected bias. CARE is particularly effective against stylistic biases such as beauty (rich content) and authority, but less effective for gender and fallacy biases, which may impact the actual quality of system answers.

| | Beauty Bias | | Fallacy Oversight Bias | | Gender Bias | | Authority Bias | |
|---|---|---|---|---|---|---|---|---|
| | MAE ($\downarrow$) | $\tau$ ($\uparrow$) | MAE ($\downarrow$) | $\tau$ ($\uparrow$) | MAE ($\downarrow$) | $\tau$ ($\uparrow$) | MAE ($\downarrow$) | $\tau$ ($\uparrow$) |
| MV | 0.9190 | 0.3336 | 1.8971 | -0.0284 | 1.7428 | 0.1272 | 0.8239 | 0.2977 |
| AVG | 0.5063 | 0.3943 | **1.4007** | **0.1181** | **1.1355** | **0.2879** | 0.3250 | 0.4288 |
| WS | 1.9225 | 0.3792 | 2.5588 | 0.0680 | 2.0217 | 0.2474 | 0.9296 | 0.4886 |
| UWS | 0.5080 | 0.4383 | 1.3826 | 0.0491 | 1.1646 | 0.2576 | 0.2705 | 0.5799 |
| CARE | **0.3749** | **0.5334** | 1.8996 | 0.0116 | 1.5985 | 0.2311 | **0.2466** | **0.6327** |

## 5.4 Comparison with Individual Intervention

**Setup.** An alternative to our confounder-aware approach is direct interventions at the individual judge level. Specifically, we compare CARE to prompt-based interventions proposed by [34], which instruct LLM judges to account for known sources of bias. The intervened prompt used for this comparison is included in Appendix E.

**Results.** Table 3 presents the results. While bias-aware prompting improves performance in most cases, CARE remains the top performer in the majority of settings, and even when not, it is competitive with the best. This suggests that CARE can effectively mitigate biases without relying on careful prompt engineering.

## 5.5 Robustness to Confounding Factors

**Setup.** We evaluate robustness using the dataset from [8], in which LLM responses are systematically altered to introduce specific biases via targeted GPT-4 prompts. The dataset includes four types of injected bias: beauty, fallacy oversight, gender, and authority. LLM judges are prompted to assign scores from 1 to 10 for each response. Robustness is assessed by comparing aggregated scores before and after bias injection, using mean absolute error (MAE) and Kendall's $\tau$. Lower MAE and higher Kendall's $\tau$ indicate better robustness under perturbation.

**Results.** Table 4 shows that CARE exhibits strong robustness to stylistic biases—such as beauty and authority—maintaining consistent rankings and score levels. In contrast, its robustness diminishes when facing biases that alter the factual or semantic content, including logical fallacies and gender-related framing.

## 5.6 Synthetic Experiments

We evaluate the performance of CARE-Tensor using simulated binary-Gaussian mixture data. Dataset details deferred to Appendix.

**Sample Complexity Result.** We investigate how the sample size $n$ influences estimation accuracy. We estimate conditional means $\hat{\mu}_{qc}$ and latent state proportions $\hat{\pi}_{qc}$ using Algorithm 2. Subsequently, we compute the posterior probabilities $P(Q = 1 \mid J)$ via the Bayesian formula-

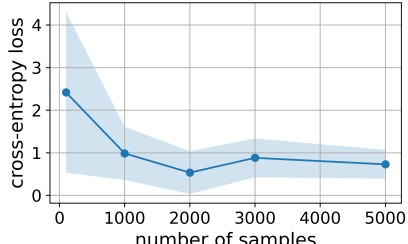

Figure 3: Averaged cross-entropy loss of our algorithm versus the number of samples. Markers denote average over three random seeds, and the shaded band denotes one standard deviation.

tion in Eq. 1. We measure the performance using cross-entropy loss. Lower entropy loss yields more accurate prediction. We observe a clear decreasing trend in cross-entropy loss as sample size increases.

**Tensor Decomposition vs SVD.** We illustrate the advantage of tensor decomposition over classical eigen-decomposition (SVD) in addressing rotation ambiguity with higher-order moments. We quantify performance using mean squared error (MSE) between true conditional means $\mu_{qc}$ and estimated means $\hat{\mu}_{qc}$. Detailed methodologies for SVD estimation are deferred to the appendix.

Evaluating across 10 random seeds, we find substantial performance differences: CARE-Tensor achieves significantly lower estimation errors with MSE ($\mathbf{0.51 \pm 0.41}$) compared to the eigen-decomposition baseline (SVD) with MSE ($\mathbf{1.18 \pm 0.74}$). This shows tensor decomposition accurately recovers conditional means without affected by rotation ambiguity.

## 6 Related Work

We discuss related work in bias in LLM-as-a-judge, label aggregation, and highlight our contribution. An extended discussion on related work can be found in Appendix B.

**Bias in LLM-as-a-judge.** Large language models (LLMs) used as automated evaluators exhibit systematic preferences such as positional, verbosity, authority, and self-enhancement biases [3, 12]. To mitigate these issues, prior work has explored prompt-based interventions [4, 35, 3] and fine-tuned evaluators such as JudgeLM and PandaLM, which aims to align model judgments more closely with human preferences [12, 5, 36]. *While effective locally, these techniques debias each single LLM judge and do not address the downstream problem of aggregating multiple, potentially correlated, LLM scores.*

**Label Aggregation.** Classic aggregation models such as Dawid–Skene [37], GLAD [38], and MACE [39] infer latent truth by modeling annotator-specific error rates. Weak-supervision frameworks generalize this idea to programmatic sources [25, 40, 26]. Recently, [2] introduce GED, a framework that ensembles and denoises preference graphs from multiple weak LLM evaluators to produce consistent and reliable model rankings. [41] analyzed various inference methods for decoding LLM-as-a-judge by looking at the judge probability distributions and computing statistics such as mean and mode (i.e greedy decoding) and studied how pre- vs post-aggregation of judge outputs affect the judge scores. *However, existing methods do not account for shared confounding factors that systematically influence annotators or LLMs alike.*

**Our Contribution.** We propose the first *confounder-aware aggregation* method for the LLM-as-a-judge setting. Unlike prior work that assumes independent annotator noise around a latent true score, our approach explicitly models shared latent confounders—such as verbosity or formality—that may jointly affect all judges. This bridges the gap between single-judge bias mitigation and statistical aggregation, enabling more reliable consensus scores in the presence of correlated judgment errors.

## 7 Conclusion

We introduce CARE, a confounder-aware aggregation framework that formulates multi-judge scoring as inference in a higher-rank latent-variable model and delivers three main contributions. **(i)** It explicitly models shared confounders, providing an aggregation scheme tailored to LLM-judge scenarios. **(ii)** It offers statistically principled estimators—sparse-plus-low-rank covariance recovery and tensor method—with provable identifiability. **(iii)** On three public benchmarks, CARE lowers MAE and raises Kendall's $\tau$ by up to 15%. Taken together, these advances enable principled, scalable, and low-cost evaluation pipelines for LLMs.

**Limitations.** Our theory assumes sufficient sparsity and approximate factor orthogonality; strong collinearity among latent variables, or latent components exhibiting similar spectral strengths may still hinder identifiability. In addition, selecting the "quality" factor currently relies on a simple loading-balance heuristic that can be unstable when confounders dominate, and our experiments are confined to English, text-only, scalar ratings—generalization to multilingual or multimodal settings remains future work.

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

The appendix is structured as follows. It starts with the glossary table, defining key notations used throughout the paper in Appendix A. Next, Appendix B discusses additional related work. In Appendix C, we introduce details about our tensor-based CARE algorithm, discussion for general CARE method, and additional discussion about method heuristics. Following this, Appendix D offers theoretical support of our approach and supported proofs. It includes the graphical model formulation, graph structure recovery error bound, sample complexity, and the misspecification error arising from incorrectly characterized confounding factors. Subsequently, Appendix E provides experimental details and additional experiment results. Finally, Appendix F concludes by discussing the broader impacts and limitations of the work.

## A  Glossary

The notations are summarized in Table 5 below.

Table 5: Glossary of variables and symbols used in this paper.

| Symbol | Definition |
| --- | --- |
| $(J_1, \ldots, J_p)$ | $p$ vector of Judges score |
| $Q$ | True-quality latent variable |
| $(C_1, \ldots, C_k)$ | $k$ latent confounder variables |
| $H$ | All the hidden variables (true + confounder) i.e $(Q\ C_1, \ldots, C_k)$ |
| $h$ | dimension of $H$ i.e all hidden variables $= k + 1$ |
| $X$ | Score matrix of dimension ($n \times p$) where $n$ is the number of examples and $p$ is the number of judges |
| $K$ | Precision matrix |
| $K_{oo}$ | Observable-observable connection matrix |
| $K_{oh}$ | Observable-latent connection matrix |
| $K_{hh}$ | Latent-latent connection matrix |
| $\Sigma_o$ | Covariance matrix of observable variables |
| $S$ | Sparse matrix ($\mathbb{R}^{p \times p}$) which encodes edges between judges |
| $L$ | Low-rank matrix (with $rank(L) \le h$) which captures dependencies mediated by latent variables |
| $R$ | Rotation matrix ($\mathbb{R}^{h \times h}$) |
| $\gamma_n$ | Regularization for sparse and low-rank matrix $S$ in Algorithm 1 |
| $\tau$ | Regularization for low-rank matrix $L$ in Algorithm 1 |
| $\hat{s}_{\mathrm{agg}}^{(i)}$ | Aggregated scores for $i$th example in the dataset from $p$ judges |
| $\hat{\Sigma}$ | Sample precision estimation or covariance matrix |
| $\hat{S}$ | Sample Sparse matrix ($\mathbb{R}^{p \times p}$) which encodes direct connectional edges among judges |
| $\hat{L}$ | Sample Low-rank matrix (with $rank(L) \le h$) which captures dependencies mediated by latent variables |
| $U$ | Latent factor extraction matrix i.e latent-judge connections ($\mathbb{R}^{p \times h}$) from Algorithm 1 |
| $\Theta$ | Precision matrix |
| $w$ | Weight for aggregating judges |
| $\lambda$ | Singular values of $L$ |
| $u^\star$ | Singular vector of $L$ corresponds to true quality factor |
| $\lambda^\star$ | Singular value of $L$ that corresponds to true quality factor |
| $\mu_{qc}$ | Conditional mean of judges given $Q = q, C = c$ |
| $\hat{\mu}_{qc}$ | Estimated conditional mean of judges given $Q = q, C = c$ |
| $\pi_{qc}$ | Probability of $Q = q, C = c$ |
| $\hat{\pi}_{qc}$ | Estimation of probability of $Q = q, C = c$ |
| $\{\mathcal{G}_\ell\}_{\ell=1}^3$ | Groups of judges that are independent of judges outside the group |
| $\hat{T}$ | Empirical 3-way tensor |
| $\hat{\mu}_{qc}^{(1)}, \hat{\mu}_{qc}^{(2)}, \hat{\mu}_{qc}^{(3)}$ | Estimated conditional mean of three views |
| $\hat{\mu}_{\rho(r)}$ | Estimated conditional mean of judges after permutation |
| $\mu_{\mathrm{anchor}(r)}$ | Conditional mean of anchor sets |

# B Extended Related Work

## B.1 Biases in LLM–as–a–Judge

Large language models (LLMs) have quickly become the standard automatic evaluators for generation tasks because they correlate well with human judgments in translation and summarization [42, 43, 44]. Yet a growing body of work shows that these models are far from impartial. **Positional bias**—preferring the *second* answer in a pairwise comparison—was first noted in MT-Bench [1] and later quantified in detail by [45], who observed reversals of up to 30% when simply swapping order. **Verbosity bias**, wherein longer answers receive higher scores regardless of quality, is highlighted by [8]. LLM judges also display **self-enhancement bias**, overrating responses produced by models from the same family [46]. Less studied but equally problematic are **concreteness/authority biases**: judges over-reward answers that contain citations, numbers, or confident tone even when these features are irrelevant [47].

Mitigation strategies span two levels. *Prompt-level interventions* randomize answer order, enforce symmetric formatting, and instruct the judge to ignore superficial features [45, 36]. Adding chain-of-thought rationales or decomposing the rubric into sub-criteria (accuracy, conciseness, style) also moderates shallow heuristics [48]. On the *model level*, fine-tuned evaluators such as JudgeLM [49] and Split-and-Merge Judge [36] are trained on curated data that explicitly counter positional and length biases. Peer-review and debate schemes go a step further: PRD lets a second LLM critique the first judge and often corrects biased decisions [50], while [48] show that dialog with a more persuasive model leads to more truthful verdicts.

Despite progress, most debiasing work treats a *single* judge in isolation. When evaluations aggregate many LLM scorers—for robustness, cost sharing, or diversity—biases can compound in complex ways that individual fixes do not capture.

## B.2 Label Aggregation for Multiple Noisy Evaluators

**Weak-supervision.** Treating each LLM prompt or model as a noisy *labeling function* aligns aggregation with modern weak supervision. Snorkel [51, 52] estimates source accuracies and dependencies to denoise programmatic labels, laying the foundation for LLM-prompt aggregation. [40] introduces a scalable moment-matching estimator with closed-form weights.[26] generalizes label models beyond categorical labels to arbitrary metric spaces, greatly expanding their applicability. [53] jointly optimizes a classifier and a differentiable label model, outperforming two-stage pipelines when sources are dependent. Firebolt further removes requirements on known class priors or source independence, estimating class-specific accuracies and correlations in closed form [54]. [55] shows that fixing source bias in labeling functions using optimal transport can improve both accuracy and fairness.

**Aggregation of multiple *LLM* judges.** Recent work shows that *ensembling smaller evaluators can beat a single large judge*. The **PoLL** jury combines three diverse 7–35B models and attains higher correlation with human ratings than GPT-4 while costing 7× less and reducing bias [56]. **GED** merges preference graphs from weak evaluators (Llama3-8B, Mistral-7B, Qwen2-7B) and denoises cycles; its DAG ranking surpasses a single 72B judge on ten benchmarks [57]. **JudgeBlender** ensembles either multiple models or multiple prompts, improving precision and consistency of relevance judgments over any individual LLM [58]. These findings echo classic "wisdom-of-crowds" results—when paired with principled aggregation, a panel of smaller, heterogeneous judges can outperform a much larger model, offering a practical path toward reliable, low-cost evaluation.

## B.3 Our Contribution in Context

Prior research either (i) debiases one judge at a time or (ii) aggregates multiple judges assuming independent noise. Our confounder-aware aggregation unifies these threads. We posit latent factors (e.g., verbosity, formality) that influence *all* judges simultaneously and show how to infer both the latent truth and the shared confounders. This yields more reliable consensus scores when individual judges—human or LLM—share systemic biases.

# C  Algorithm Details

This section details the implementation of our CARE framework. Specifically, it includes the full CARE tensor algorithm, details about SVD baseline method for comparing our tensor-based algorithm, generalizations beyond Gaussian assumptions, and practical heuristics to address non-orthogonality in latent factors and justification for where the sparse structure lies in mixed Gaussian data.

## C.1  Tensor-based CARE Algorithm

---

**Algorithm 2** CARE (T)

---

**Input:** Score matrix $J \in \mathbb{R}^{n \times p}$, tolerance $\varepsilon$.
**Output:** Estimates $\{\hat{\mu}_{qc}, \hat{\pi}_{qc}\}_{q,c \in \{0,1\}}$.

    **A. Anchor discovery (graph partition)**
1: Compute the sample covariance $\hat{\Sigma} = J^\top J / n$ and perform the sparse+low-rank split $\hat{\Sigma} \approx \hat{S} + \hat{L}$ (Alg. 1).
2: Partition judges into three disjoint groups $\{\mathcal{G}_\ell\}_{\ell=1}^3$ that satisfy

$$a \neq b, \ j_1 \in \mathcal{G}_a, \ j_2 \in \mathcal{G}_b \implies |\hat{S}_{j_1, j_2}| \leq \varepsilon,$$

    ensuring no direct edges with strength greater than $\epsilon$ can exist across groups.

    **B. Empirical third-order moment tensor**
3: **for** $\ell = 1, 2, 3$ **do**
4:     $X_\ell \leftarrow$ columns of $J$ indexed by $\mathcal{G}_\ell$                             $\triangleright X_\ell \in \mathbb{R}^{n \times |\mathcal{G}_\ell|}$
5: **end for**
6: Compute

$$\hat{T} = \frac{1}{n} \sum_{i=1}^n X_1^{(i)} \otimes X_2^{(i)} \otimes X_3^{(i)} \ \in \ \mathbb{R}^{|\mathcal{G}_1| \times |\mathcal{G}_2| \times |\mathcal{G}_3|}.$$

    **C. Tensor decomposition**
7: Run a CP tensor-power decomposition on $\hat{T}$ to obtain $k = 4$ components
    $\left\{(\hat{\pi}_{qc}, \hat{\mu}_{qc}^{(1)}, \hat{\mu}_{qc}^{(2)}, \hat{\mu}_{qc}^{(3)})\right\}_{q,c \in \{0,1\}^2}$, where $\hat{\pi}_{qc} > 0$ and $\hat{\mu}_{qc}^{(\ell)} \in \mathbb{R}^{|\mathcal{G}_\ell|}$.
    **D. Assemble full means**
8: **for** $q, c \in \{0,1\}^2$ **do**
9:     $\hat{\mu}_{qc} \leftarrow \text{concat}(\hat{\mu}_{qc}^{(1)}, \hat{\mu}_{qc}^{(2)}, \hat{\mu}_{qc}^{(3)}) \in \mathbb{R}^p$.
10: **end for**
    **E. State alignment with anchors**
11: Find the permutation $\rho$ of $\{1, \ldots, 4\}$ that minimizes $\sum_{r=1}^4 \left\| \hat{\mu}_{\rho(r)} - \mu_{\text{anchor}(r)} \right\|_2^2$, where the four anchor prototypes correspond to $(Q, C) = \{00, 01, 10, 11\}$.
12: $(\hat{\mu}_{00}, \hat{\mu}_{01}, \hat{\mu}_{10}, \hat{\mu}_{11}) \leftarrow (\hat{\mu}_{\rho(1)}, \hat{\mu}_{\rho(2)}, \hat{\mu}_{\rho(3)}, \hat{\mu}_{\rho(4)})$.
    **F. Mixing weights**
13: $(\hat{\pi}_{00}, \hat{\pi}_{01}, \hat{\pi}_{10}, \hat{\pi}_{11}) \leftarrow (\hat{\pi}_{\rho(1)}, \hat{\pi}_{\rho(2)}, \hat{\pi}_{\rho(3)}, \hat{\pi}_{\rho(4)})$.
14: **return** $\{\hat{\mu}_{qc}, \hat{\pi}_{qc}\}_{q,c \in \{0,1\}}$.

---

## C.2  SVD Baseline in Synthetic Experiment

We form the empirical two-way moment between view 1 and view 2:

$$\widehat{M}_{1,2} \ = \ \frac{1}{n} \sum_{i=1}^n X_1^{(i)} X_2^{(i)\top} \ = \ \sum_{q,c} \pi_{q,c} \, \mu_{1,q,c} \, \mu_{2,q,c}^\top \ + \ \text{sampling noise},$$

where $\pi_{q,c} = \Pr[Q = q, C = c]$ and $\mu_{v,q,c} = E[J_v \mid Q = q, C = c]$ for judge/view $v$ A singular-value decomposition

$$\widehat{M}_{1,2} \ = \ U_{12} \, \Sigma_{12} \, V_{12}^\top$$

yields factor matrices

$$U_{12} \, \Sigma_{12}^{1/2} \approx [\mu_{1,q,c}] \, R, \quad V_{12} \, \Sigma_{12}^{1/2} \approx [\mu_{2,q,c}] \, R,$$

where $R \in O(4)$ is an unknown orthogonal matrix.

Repeating on $\widehat{M}_{1,3} = \frac{1}{n} \sum_i X_1^{(i)} X_3^{(i)\top} = U_{13} \, \Sigma_{13} \, V_{13}^\top$ produces a second rotated copy of $[\mu_{1,q,c}]$. We solve the Procrustes problem

$$R = \arg \min_{O \in O(4)} \left\| U_{12} \, \Sigma_{12}^{1/2} - U_{13} \, \Sigma_{13}^{1/2} \, O \right\| * F,$$

then set $\hat{\mu}_{2,q,c} = (V_{12} \, \Sigma_{12}^{1/2}) \, R^\top$ and $\hat{\mu}_{3,q,c} = (V_{13} \, \Sigma_{13}^{1/2}) \, R^\top$ to align all three views.

This SVD baseline recovers $\{\mu_{v,q,c}\}$ up to the permutation/sign ambiguity inherent in any orthogonal transform.

## C.3 Genral CARE Setup

**Extension Beyond the Gaussian Observation Model.** The multivariate-Gaussian assumption for $J|H$ is convenient—its first two or three moments already encode all information needed for the sparse + low-rank and tensor steps—but it is not a requirement. Because CARE learns the *graphical* structure, the same pipeline applies whenever each judge's conditional distribution lies in an exponential family or, more generally, a latent-variable generalized linear model (GLM):

- **Categorical or ordinal scores.** For Likert ratings or pairwise preferences we can set

$$J_i \mid H \sim \mathrm{Categorical}\big(\mathrm{softmax}(W_i^\top H)\big) \quad \text{or} \quad \mathrm{Ordinal-logit}(W_i^\top H).$$

  The graph—hence the sparse mask $S$—is unchanged; only the node-wise likelihoods differ. We still recover $S$ from conditional-mutual-information or pseudo-likelihood scores, and we still factorize higher-order indicator moments such as $\mathbb{E}\big[\mathbf{1}_{\{J_a=\ell\}} \, \mathbf{1}_{\{J_b=m\}} \, \mathbf{1}_{\{J_c=n\}}\big]$.

- **Mixed Discrete-Continous Scores.** When some judges output real scores and others categorical flags, we use a mixed conditional distribution:

$$p(J|H) = \big[\Pi_{i \in \mathrm{Cont.}} \mathcal{N}(J_i; \mu_{H_i}, \sigma_i^2)\big] \big[\Pi_{j \in \mathrm{Disc.}} \mathrm{Bernoulli}(\sigma(W_j^\top H))\big].$$

  CARE forms mixed raw/indicator moments, and identifiability again follows from standard tensor-decomposition guarantees for mixed conditional means.

- **Heavy-tailed or skewed real scores.** When numeric scores are skewed or contain outliers, a multivariate-$t$ or Gaussian scale mixture is appropriate. Up to a scalar factor, the covariance still decomposes as sparse + low-rank, so Steps 1–2 of Algorithm 1 work after a simple rescaling.

Empirically, we find that replacing the Gaussian local likelihood only affects the estimation of sparse structure and extraction of latent factors, not the subsequent symmetry-breaking or posterior computation; thus the overall CARE pipeline generalizes with minimal adjustments.

## C.4 Heuristics and Justifications

**Heuristic for Addressing Orthogonality Violations in CARE (SVD).**

Existing heuristics for identifying the true quality latent factor can estimate corresponding weights, but they often suffer from bias when the orthogonality assumption—central to the application of SVD—is violated. This issue commonly arises in real-world datasets. We found the following weighting rule effective in both synthetic and real-world settings:

$$w \leftarrow \lambda^\star u^\star - \sum_{u_i \in U \setminus \{u^\star\}} \lambda_i u_i,$$

where $w$ represents the learned weights for each judge, $\lambda^*$ and $u^*$ is the singular value and vector of $L$ that corresponds to the direction that is closest to true quality latent variable, $\lambda_i, u_i$ represent rest of the singular values and vectors, which can be interpreted as spurious/confounding factors.

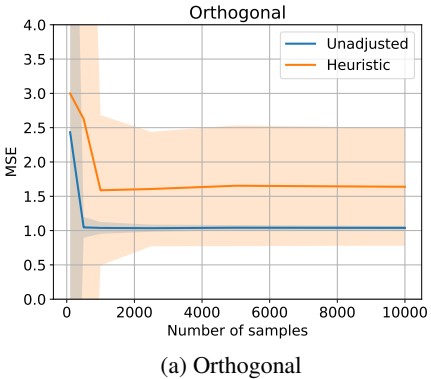

(a) Orthogonal

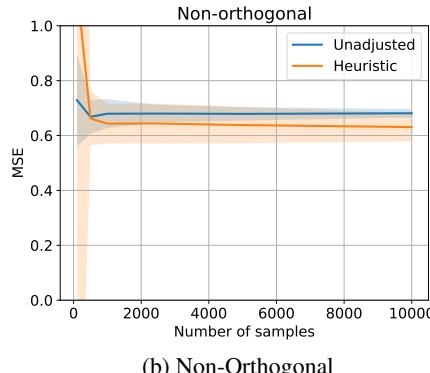

(b) Non-Orthogonal

Figure 4: Effect of the proposed heuristic in a fully Gaussian synthetic setup. We estimate the true quality variable $Q$ and report the mean squared error. The heuristic improves estimation in the non-orthogonal setting, but slightly degrades performance in the orthogonal setting where true and confounding components are disjoint.

This rule intuitively subtracts the influence of overlapping (non-orthogonal) confounding components from the estimated true score factor.

Figure 4 illustrates the effect of this heuristic in a synthetic fully Gaussian setup. In the non-orthogonal case—where confounding components overlap with the true signal—the heuristic improves the estimation of the true latent variable. In contrast, it underperforms in the orthogonal case, where judges influenced by true scores are cleanly separated from those influenced by confounders.

**Justification of Decomposing Covariance Matrix.** In the joint-Gaussian setting we decompose the *precision* matrix, whose sparsity pattern directly encodes conditional independences in an undirected graphical model. For a *mixed* Gaussian model, however, each observation $J \in \mathbb{R}^p$ is generated by first drawing a latent class label $Q, C \in \{0, 1\}^2$ (with probabilities $\pi_{qc}$) and then sampling

$$J \mid Q, C = q, c \sim \mathcal{N}\big(\mu_{qc}, \Sigma\big),$$

where the within-component covariance $\Sigma$ does not depend on $q, c$. Because the latent variable only perturbs the mean, the marginal covariance of $J$ splits, via the *law of total covariance*, into

$$\text{Cov}(J) = \underbrace{\mathbb{E}\big[\text{Cov}(J \mid Q, C)\big]}_{=\Sigma} + \underbrace{\text{Cov}\big(\mathbb{E}[J \mid Q, C]\big)}_{=\sum_{q,c} \pi_{qc} (\mu_{qc} - \bar{\mu})(\mu_{qc} - \bar{\mu})^\top} \quad , \quad \bar{\mu} := \sum_{q,c} \pi_{qc} \mu_{qc}. \quad (2)$$

The first term, $\Sigma$, is the same sparse block-diagonal matrix we plant in the simulator to model direct judge–judge interactions; the second term is an outer-product mixture of at most $4$ linearly independent directions and hence has rank $\leq 4$. Equation equation 2 therefore exhibits the population covariance as a *sparse + low-rank* decomposition,

$$\text{Cov}(J) = S + L, \quad S = \Sigma \text{ (sparse)}, \quad L = \text{Cov}\big(\mathbb{E}[J \mid Q, C]\big) \text{ (low rank)}.$$

Because sparsity now lives in $S$, not in the inverse covariance, estimating $S$ and $L$ by fitting a sparse-plus-low-rank model directly to the empirical covariance is both natural and statistically identifiable for the mixed Gaussian case.

# D  Theory

We formalize the graphical model under joint gaussian distribution and notation (Section D.1), then discuss the identifiability of graph structure with exact and approximate recovery (Section D.2) and quantify the sample complexity required for consistent recovery of our SVD-based algorithm (Section D.3). Next, we present the model misspecification error when confounding factor is not correctly characterized (Section D.4). Finally, we discuss sample complexity required for tensor-based algorithm under mixed Gaussian distribution (Section D.5. All proofs are included in Section D.6.

## D.1  Model and Notation

We discuss the model under joint-gaussian distribution where all variables follow the same definitions as in Section 3. Briefly, $J = (J_1, \ldots, J_p)^\top$ stacks the $p$ observable judge scores, and $H = (Q, C_1, \ldots, C_k)^\top$ collects the $h = k + 1$ latent variables.

$$\Sigma = \mathrm{Cov}\big[(J, H)^\top\big], \qquad \Sigma^{-1} = K = \begin{pmatrix} K_{JJ} & K_{JH} \\ K_{HJ} & K_{HH} \end{pmatrix},$$

where the subscript $J$ (resp. $H$) refers to observable (resp. latent) coordinates.

The observable block factorizes via the Schur complement:

$$(\Sigma_{JJ})^{-1} = S + L, \quad S = K_{JJ}, \quad L = K_{JH} K_{HH}^{-1} K_{HJ}.$$

Here $\Sigma_o$ is the covariance matrix of observable variables, $S \in \mathbb{R}^{p \times p}$ is sparse and encodes direct conditional edges among judges, $L$ is low-rank with $\mathrm{rank}(L) \leq h$ and captures dependencies mediated by the latent variables. Entry $(K_{JH})_{i\ell}$ is the edge weight between judge $i$ and latent factor $\ell$.

## D.2  Graph Structure Identifiability

While $(S, L)$ can be recovered (e.g. via convex sparse-plus-low-rank regularization [18], the finer structure of $K_{JH}$ is usually not identifiable from $L$. For example, for arbitrary rotation matrix $R \in \mathbb{R}^{h \times h}$, $L = (K_{JH} K_{HH}^{-1/2} R)(R^\top K_{HH}^{-1/2} K_{HJ})$, this indicates one cannot distinguish $K_{JH} K_{HH}^{-1/2}$ from $K_{JH} K_{HH}^{-1/2} R$ without further constraints. Hence, we need to impose additional assumptions:

**Assumption D.1** (Latent–latent independence and eigen-gap). $K_{HH} = \mathrm{diag}(d_1, \ldots, d_h)$ with $d_1 > d_2 > \cdots > d_h > 0$.

**Assumption D.2** (Orthogonal latent–observable connections). The columns of $K_{JH}$ are orthogonal, i.e. $K_{JH}^\top K_{JH}$ is diagonal. A special case is the *disjoint-support* model where each judge connects to exactly one latent factor.

Next, we provide an exact recovery result given the above assumptions.

**Theorem D.3** (Exact Recovery). *Under Assumptions 1 and 2, columns in $K_{JH}$ are identifiable up to column permutations and sign flips.*

Real-world data rarely satisfy the exact orthogonality in Assumption D.2. To assess robustness, consider the following perturbed connection matrix:

$$\tilde{K}_{JH} = K_{JH} + E, \qquad \|E\|_2 \text{ small}.$$

The associated low-rank part is $\tilde{L} = \tilde{K}_{JH} K_{HH}^{-1} \tilde{K}_{HJ}$. Let the eigen-pairs of $L = K_{JH} K_{HH}^{-1} K_{HJ}$ and $\tilde{L}$ be $\{(\lambda_i, u_i)\}_{i=1}^h$ and $\{(\tilde{\lambda}_i, \tilde{u}_i)\}_{i=1}^h$, ordered so that $\lambda_1 > \cdots > \lambda_h > 0$, and denote the eigen-gap by

$$\delta_i = \min_{j \neq i} |\lambda_i - \lambda_j| > 0.$$

**Theorem D.4** (Stability under approximate orthogonality). *For every $i \in [h]$,*

$$\|\hat{u}_i - u_i\|_2 \ \leq \ \frac{2\|K_{HH}^{-1}\|_2 \, \|E\|_2}{\delta_i} \ + \ O\big(\|E\|_2^2\big).$$

This indicates that latent–observable directions remain identifiable (up to column permutations and sign flips) whenever the perturbation norm $\|E\|_2$ is small relative to the eigen-gap $\delta_i$. We defer the proof to Appendix D.6.

## D.3 Sample Complexity Bound

We now quantify how many i.i.d. samples are needed for the two–stage estimator in Algorithm 1 to recover the latent–observable directions $K_{JH} \in \mathbb{R}^{p \times h}$.

As detailed in Algorithm 1, our estimator for $K_{JH}$ proceeds in two stages: first, a sparse + low-rank decomposition of sample precision matrix. Second, we extract the latent–observable directions by taking the rank-$h$ eigen-decomposition $\hat{L}_n = \sum_{i=1}^{h} \hat{\lambda}_i \hat{u}_i \hat{u}_i^\top$ and setting $\hat{K}_{JH} := [\hat{u}_1, \dots, \hat{u}_h]$.

**Theorem D.5** (Sample complexity for recovering $K_{JH}$). *Let $L^* = K_{JH} K_{HH}^{-1} K_{HJ} \in \mathbb{R}^{p \times p}$ have distinct eigenvalues $\lambda_1 > \cdots > \lambda_h$ and define the (global) eigengap $\delta := \min_{1 \le i < j \le h} |\lambda_i - \lambda_j|$. Assume the identifiability, incoherence, and curvature conditions of [18]. Then for any $\epsilon > 0$, with probability at least $1 - 2e^{-\epsilon}$,*

$$\max_{i \le h} \| \hat{u}_i - u_i \|_2 = O\Big(\frac{\sqrt{\epsilon}}{\sqrt{n}\,\xi(T)\,\delta}\Big),$$

*where $n$ is the sample size, $\hat{u}_i$ and $u_i$ are the $i$-th eigenvectors of $\hat{L}_n$ and $L^*$ respectively. $T = T(L^*)$ is the tangent space of $L^*$, $\xi(T)$ is the curvature constant from [18].*

We defer the proof to Appendix D.6. At a high-level, we adapt the identifiability, incoherence and curvature conditions from Theorem 4.1 of [18] and combine it with extended result of Davis-Khan's theorem [59].

This bound shows that the column-wise $\ell_2$ error decays at the standard parametric rate $n^{-1/2}$, and is attenuated by both the manifold curvature $\xi(T)$ and the eigengap $\delta$. Achieving an accuracy of at most $\alpha \in (0, 1)$ therefore requires

$$n = \tilde{O}\Big(\frac{\epsilon}{\xi(T)^2 \delta^2 \alpha^2}\Big)$$

samples, up to universal constants and log-factors.

## D.4 Misspecification Error

Many label aggregation frameworks (e.g.,[25, 40, 26]) assume a *single* latent variable that explains the observed labels. However, in setups like LLM-as-a-judge, the scores may be influenced by additional latent factors or confounders that also affect the observed annotations. Ignoring these *confounder* latents leads to model misspecification, which can bias the aggregated labels. We characterize this bias and analyze its impact on the estimated aggregation weights.

Let $L^* = \sum_{\ell=1}^{h} \frac{1}{d_\ell} \mathbf{k}_\ell \mathbf{k}_\ell^T$ be the true rank-$h$ low-rank component of the observable precision matrix, derived from the latent-observable connection matrix $K_{JH} = [\mathbf{k}_1, \dots, \mathbf{k}_h]$ and latent-latent precision $K_{HH} = \mathrm{diag}(d_1, \dots, d_h)$. Let $\mathbf{u}_1^{\text{true}} = \mathbf{k}_1 / \|\mathbf{k}_1\|_2$ be the true direction of influence for the quality score latent variable $Q$ (assuming $\mathbf{k}_1 \ne \mathbf{0}$).

Define $\mathbf{A} = \frac{1}{d_1} \mathbf{k}_1 \mathbf{k}_1^T$. Its principal (and only non-zero) eigenvalue is $\lambda_1 = \frac{1}{d_1} \|\mathbf{k}_1\|_2^2$, and its spectral gap (to its other zero eigenvalues) is $\delta = \lambda_1$. Let $\mathbf{E} = \sum_{\ell=2}^{h} \frac{1}{d_\ell} \mathbf{k}_\ell \mathbf{k}_\ell^T$ be the confounding component, so $L^* = \mathbf{A} + \mathbf{E}$. Let $\mathbf{v}_1$ be the principal unit-norm eigenvector of $L^*$. When a rank-1 model is fitted, the estimated direction is $\hat{\mathbf{u}}_1^{\text{pop}} = \mathbf{v}_1$.

**Theorem D.6.** *If $\|\mathbf{E}\|_{op} \le \delta/2$, the $\ell_2$ deviation of the estimated direction $\mathbf{v}_1$ from $\mathbf{u}_1^{\text{true}}$ is bounded by:*

$$\big\|\mathbf{v}_1 - s\mathbf{u}_1^{\text{true}}\big\|_2 \le \frac{2\,\|\mathbf{E}\|_{op}}{\delta} = \frac{2\Big\|\sum_{\ell=2}^{h} \frac{1}{d_\ell} \mathbf{k}_\ell \mathbf{k}_\ell^T\Big\|_{op}}{\frac{1}{d_1}\|\mathbf{k}_1\|_2^2}$$

*for a sign $s = \pm 1$ (chosen so that $s(\mathbf{u}_1^{\text{true}})^T \mathbf{v}_1 \ge 0$).*

*Proof.* By Davis-Kahan theorem (Theorem 2 in [59]), if $\|\mathbf{E}\|_{op} \le \delta/2$, then the $\ell_2$ distance between $\mathbf{v}_1$ and $\mathbf{u}_1^{\text{true}}$ (after aligning their signs via $s = \pm 1$) is bounded by:

$$\big\|\mathbf{v}_1 - s \cdot \mathbf{u}_1^{\text{true}}\big\|_2 \le \frac{2\,\|\mathbf{E}\|_{op}}{\delta}.$$

Plugging in $E$ yields the desired result:

$$\left|\left|\mathbf{v}_1 - s \cdot \mathbf{u}_1^{\text{true}}\right|\right|_2 \leq \frac{2\left|\left|\sum_{\ell=2}^{h} \frac{1}{d_\ell}\mathbf{k}_\ell \mathbf{k}_\ell^T\right|\right|_{\text{op}}}{\frac{1}{d_1}\left|\left|\mathbf{k}_1\right|\right|_2^2}.$$

$\square$

The theorem quantifies the directional bias in the estimated influence of $Q$ when confounders are ignored. This bias is proportional to the collective "strength" of confounders in the precision domain (numerator) and inversely proportional to $Q$'s own "strength" (denominator). Fitting a rank-1 model forces this bias, while a higher-rank model offers the capacity to separate these influences.

**Corollary D.7** (Error Bound for Estimated Conditional Mean of $Q$). *Denote the true conditional mean of true quality score latent variable $Q$ given the observable variables $O = (J_1, ..., J_p)$ be denoted by $\mathbb{E}[Q|O]_{\text{true}}$. Then, $\mathbb{E}[Q|\mathbf{o}]_{\text{true}} = -\frac{||\mathbf{k}_1||_2}{d_1}(\mathbf{u}_1^{\text{true}})^T\mathbf{o}$. Let an estimated conditional mean with the misspecified direction, $\mathbb{E}[Q|\mathbf{o}]_{\text{mis}}$, be formed using the misspecified direction $\mathbf{v}_1$ be $\mathbb{E}[Q|\mathbf{o}]_{\text{mis}} = -\frac{||\mathbf{k}_1||_2}{d_1}(s \cdot \mathbf{v}_1)^T\mathbf{o}$, where $s = \pm 1$ is chosen such that $s \cdot (\mathbf{u}_1^{\text{true}})^T\mathbf{v}_1 \geq 0$. Then, the absolute error in the estimated conditional mean due to the directional misspecification is bounded by:*

$$\left|\mathbb{E}[Q|\mathbf{o}]_{\text{mis}} - \mathbb{E}[Q|\mathbf{o}]_{\text{true}}\right| \leq \frac{2\left|\left|\sum_{\ell=2}^{h}\frac{1}{d_\ell}\mathbf{k}_\ell\mathbf{k}_\ell^T\right|\right|_{op}}{||\mathbf{k}_1||_2}\,||\mathbf{o}||_2$$

*This holds if the condition from the main theorem, $||\mathbf{E}||_{op} \leq \delta/2 = \frac{1}{2d_1}||\mathbf{k}_1||_2^2$, is met, where $\mathbf{E} = \sum_{\ell=2}^{h}\frac{1}{d_\ell}\mathbf{k}_\ell\mathbf{k}_\ell^T$.*

*Proof.* The absolute difference is:

$$\left|\mathbb{E}[Q|\mathbf{o}]_{\text{mis}} - \mathbb{E}[Q|\mathbf{o}]_{\text{true}}\right| = \left|-\frac{||\mathbf{k}_1||_2}{d_1}(s \cdot \mathbf{v}_1)^T\mathbf{o} - \left(-\frac{||\mathbf{k}_1||_2}{d_1}(\mathbf{u}_1^{\text{true}})^T\mathbf{o}\right)\right|$$

$$= \left|-\frac{||\mathbf{k}_1||_2}{d_1}(s \cdot \mathbf{v}_1 - \mathbf{u}_1^{\text{true}})^T\mathbf{o}\right|$$

$$= \frac{||\mathbf{k}_1||_2}{d_1}\left|(s \cdot \mathbf{v}_1 - \mathbf{u}_1^{\text{true}})^T\mathbf{o}\right|$$

By the Cauchy-Schwarz inequality, $\left|(\mathbf{x})^T\mathbf{y}\right| \leq ||\mathbf{x}||_2\,||\mathbf{y}||_2$. Applying this:

$$\left|\mathbb{E}[Q|\mathbf{o}]_{\text{mis}} - \mathbb{E}[Q|\mathbf{o}]_{\text{true}}\right| \leq \frac{||\mathbf{k}_1||_2}{d_1}\left|\left|s \cdot \mathbf{v}_1 - \mathbf{u}_1^{\text{true}}\right|\right|_2\,||\mathbf{o}||_2$$

The term $||s \cdot \mathbf{v}_1 - \mathbf{u}_1^{\text{true}}||_2$ is equivalent to $||\mathbf{v}_1 - s \cdot \mathbf{u}_1^{\text{true}}||_2$ from the main theorem statement, where $s$ aligns $\mathbf{u}_1^{\text{true}}$ with $\mathbf{v}_1$. From the preceding Theorem, we have the bound (where $\delta = \frac{1}{d_1}||\mathbf{k}_1||_2^2$):

$$\left|\left|\mathbf{v}_1 - s \cdot \mathbf{u}_1^{\text{true}}\right|\right|_2 \leq \frac{2||\mathbf{E}||_{\text{op}}}{\delta} = \frac{2\left|\left|\sum_{\ell=2}^{h}\frac{1}{d_\ell}\mathbf{k}_\ell\mathbf{k}_\ell^T\right|\right|_{\text{op}}}{\frac{1}{d_1}||\mathbf{k}_1||_2^2}$$

Substituting this bound into the inequality for the error in the conditional mean:

$$\left|\mathbb{E}[Q|\mathbf{o}]_{\text{mis}} - \mathbb{E}[Q|\mathbf{o}]_{\text{true}}\right| \leq \frac{||\mathbf{k}_1||_2}{d_1}\left(\frac{2||\mathbf{E}||_{\text{op}}}{\frac{1}{d_1}||\mathbf{k}_1||_2^2}\right)||\mathbf{o}||_2$$

$$= \frac{||\mathbf{k}_1||_2}{d_1} \cdot \frac{2d_1||\mathbf{E}||_{\text{op}}}{||\mathbf{k}_1||_2^2} \cdot ||\mathbf{o}||_2$$

$$= \frac{2||\mathbf{E}||_{\text{op}}}{||\mathbf{k}_1||_2}||\mathbf{o}||_2$$

$$= \frac{2\left|\left|\sum_{\ell=2}^{h}\frac{1}{d_\ell}\mathbf{k}_\ell\mathbf{k}_\ell^T\right|\right|_{\text{op}}}{||\mathbf{k}_1||_2}||\mathbf{o}||_2$$

$\square$

This corollary shows that the error in the estimated conditional mean of $Q$ (due to using the misspecified direction for $Q$'s influence) scales with:

- The magnitude of the observable vector $\mathbf{o}$ (specifically, $\|\mathbf{o}\|_2$).
- The collective strength of the confounding latent variables in the precision domain ($\left\|\sum_{\ell=2}^{h} \frac{1}{d_\ell} \mathbf{k}_\ell \mathbf{k}_\ell^T\right\|_{\text{op}}$).
- Inversely with the $\ell_2$-norm of the true connection weights of $Q$ ($\|\mathbf{k}_1\|_2$).

Especially, we see that strong confounders widen the gap bound, whereas heavier connection weights to the true score shrink it. Put differently, *misspecification hurts most when confounders are strong and the quality signal is weak.*

## D.5 Sample Complexity for CARE tensor algorithm

**Assumption D.8** (Model and identifiability). Let $J = (X_1^\top, X_2^\top, X_3^\top)^\top \in \mathbb{R}^p$ ($p = p_1 + p_2 + p_3$) be one observations i.i.d generated as

$$(Q, C) \sim \text{Multinomial}(\{\pi_{qc}\}_{q,c \in \{0,1\}}), \qquad X_\ell \mid (Q = q, C = c) \sim \mathcal{N}\big(\mu_{qc}^{(\ell)}, \Sigma\big),$$

with $\ell \in \{1, 2, 3\}$. Write $r \in [4] \leftrightarrow (q, c) \in \{0, 1\}^2$ and define $w_r := \pi_{qc}$, $a_r := \mu_{qc}^{(1)} \in \mathbb{R}^{p_1}$, $b_r := \mu_{qc}^{(2)} \in \mathbb{R}^{p_2}$, $c_r := \mu_{qc}^{(3)} \in \mathbb{R}^{p_3}$.

(A1) **Block-conditional independence.** $X_1 \perp X_2 \perp X_3 \mid (Q, C)$.

(A2) **Full-rank moment tensor.** The population third-order moment $M := \mathbb{E}[X_1 \otimes X_2 \otimes X_3] = \sum_{r=1}^{4} w_r \, a_r \otimes b_r \otimes c_r$ has rank 4, with $\pi_{\min} := \min_r \pi_r > 0$ and $\lambda_{\min} := \min_r \|a_r\|_2 \|b_r\|_2 \|c_r\|_2 > 0$.

(A3) **Non-degenerate covariance.** $\sigma_{\max}^2 := \|\Sigma\|_{\text{op}} < \infty$.

(A4) **Spectral gap.** The CP factors are uniquely defined up to scaling/sign and satisfy the eigenvalue-gap condition of Theorem 5.1 in [19]. Denote that gap by $\delta > 0$.

(A5) **Correct graph partition.** There exist a graph partition such that judges between different groups are conditional independent. Step A of Algorithm 2 returns the true groups $\mathcal{G}_1, \mathcal{G}_2, \mathcal{G}_3$.

**Theorem D.9** (Sample complexity of CARE tensor step). *Fix $0 < \varepsilon < 1$ and let the assumptions above hold. Run Algorithm 2 (CARE) on $n$ i.i.d. samples to obtain $\{\hat{\mu}_{qc}, \hat{\pi}_{qc}\}_{q,c \in \{0,1\}}$. Under Assumption D.8, there exist universal constants $C_1, C_2 > 0$ such that if*

$$n \geq C_1 \, \frac{\sigma_{\max}^6}{\delta^2 \, \pi_{\min}^2} \, p \, \log(p/\varepsilon),$$

*then with probability at least $1 - \varepsilon$*

$$\max_{q,c} \left\|\hat{\mu}_{qc} - \mu_{qc}\right\|_2 \leq C_1 \frac{\sigma_{\max}^3}{\delta} \sqrt{\frac{p \, \log(p/\varepsilon)}{n}}, \qquad \max_{q,c} \left|\hat{\pi}_{qc} - \pi_{qc}\right| \leq C_2 \sqrt{\frac{p \, \log(p/\varepsilon)}{n}}.$$

We defer the proof to D.6.

## D.6 Proofs

**Proof of Theorem D.3**

*Proof.* Let low-rank matrix satisfies $L = \sum_{i=1}^{h} d_i \, u_i u_i^\top$ with $u_i$ the $i$-th column of $K_{oh}$. By Assumption D.2 the $u_i$ are mutually orthogonal, and by Assumption D.1 the eigenvalues $d_1 > \cdots > d_h$ are distinct; hence this rank-1 decomposition is the (unique) spectral decomposition of $L$. Thus each $u_i$ is identifiable from $L$ up to sign and ordering, proving the theorem. $\qquad\square$

 **Proof of Theorem D.4**

 *Proof.* We apply standard matrix perturbation theory for eigenvectors. Starting from the eigenvalue
 decomposition:
$$L\, u_i = \lambda_i\, u_i,$$

 we write the perturbed matrix as

$$\tilde{L} = (K_{oh} + E)K_{hh}^{-1}(K_{oh} + E)^{\top} = L \; + \; K_{oh}K_{hh}^{-1}E^{\top} \; + \; EK_{hh}^{-1}K_{oh}^{\top} \; + \; EK_{hh}^{-1}E^{\top}.$$

 Let $\Delta L = \tilde{L} - L$. By the Davis–Kahan theorem,

$$\|\hat{u}_i - u_i\|_2 \; \leq \; \frac{2\,\|\Delta L\|_2}{\delta_i},$$

 where $\delta_i = \min_{j \neq i} |\lambda_i - \lambda_j| > 0$. Moreover,

$$\|\Delta L\|_2 \; \leq \; 2\,\|K_{oh}\|_2\,\|K_{hh}^{-1}\|_2\,\|E\|_2 \; + \; O(\|E\|_2^2)$$

 and $\|K_{oh}\|_2 = 1$. Hence

$$\|\hat{u}_i - u_i\|_2 \; \leq \; \frac{2\,\|K_{hh}^{-1}\|_2\,\|E\|_2}{\delta_i} \; + \; O(\|E\|_2^2).$$

 This completes the proof. $\qquad\square$

 **Proof of Theorem D.5**

 *Proof of Theorem D.5.* **Step 1 – Spectral error of $\hat{L}_n$.** Apply Chandrasekaran et al.'s Theorem 4.1
 with the regularization parameters

$$\gamma_n \; = \; \frac{48\sqrt{2}D\psi(2-\nu)}{\xi(T)\nu}\sqrt{\frac{\epsilon}{n}}, \qquad \sigma, \theta \text{ as in their conditions (3)–(4).}$$

 Under the incoherence and curvature conditions of their Proposition 3.3, there exists a universal
 constant $C_1 > 0$ such that, with probability at least $1 - 2e^{-\epsilon}$,

$$\big\|\,\hat{L}_n - L^*\big\|_2 \; \leq \; C_1\,\frac{\sqrt{\epsilon/n}}{\xi(T)}. \tag{3}$$

 **Step 2 – Eigenvector perturbation via Davis–Kahan.** Let $L^* = U\Lambda U^{\top}$ with $\Lambda =$
 $\mathrm{diag}(\lambda_1, \ldots, \lambda_h, 0, \ldots, 0)$ and collect the top–$h$ eigenvectors in $U_h = [u_1, \ldots, u_h]$. Write the
 spectral decomposition of the estimator as $\hat{L}_n = \hat{U}_h\hat{\Lambda}\hat{U}_h^{\top} + R$, where $R$ contains only the eigen-
 components of rank $> h$. Set the perturbation $E := \hat{L}_n - L^*$.

 Applying Corollary 3 from [59] to the $i$-th eigenpair gives

$$\|u_i - \hat{u}_i\|_2 \; \leq \; \frac{2^{3/2}\|E\|_2}{\delta_i}. \tag{4}$$

 **Step 3 – Combine the two bounds.** Insert equation 3 into equation 4:

$$\|\,\hat{u}_i - u_i\,\|_2 \; \leq \; \frac{2^{3/2}C_1}{\delta\,\xi(T)}\sqrt{\frac{\epsilon}{n}} \qquad \forall\, i \in [h],$$

 and take the maximum over $i$. This proves the advertised high-probability bound

$$\max_{i \leq h} \|\,\hat{u}_i - u_i\,\|_2 \; = \; O\Big(\tfrac{\sqrt{\epsilon/n}}{\xi(T)\,\delta}\Big).$$

 **Step 4 – Invert to a sample-size requirement.** Setting the right-hand side to a target accuracy
 $\varepsilon \in (0, 1)$ and solving for $n$ yields $n \; \geq \; \frac{4C_1^2}{\varepsilon^2}\frac{\epsilon}{\xi(T)^2\delta^2}$, which is the sample-complexity statement in
 the theorem. $\qquad\square$

**Proof for Theorem D.9**

*Proof sketch.* **Step 1: Concentration of the empirical tensor.** Let $\hat{M} := \frac{1}{n} \sum_{i=1}^{n} X_1^{(i)} \otimes X_2^{(i)} \otimes X_3^{(i)}$. Because each $X_\ell$ is sub-Gaussian with proxy $\sigma_{\max}$, the operator-norm Bernstein bound for order-3 tensors (Lemma 5 of 60) yields

$$\|\hat{M} - M\|_{\mathrm{op}} = O\Big(\sigma_{\max}^3 \sqrt{\tfrac{p \log(p/\varepsilon)}{n}}\Big) \quad \text{w.p. } 1 - \varepsilon/2.$$

**Step 2: Robust CP decomposition.** Applying the non-symmetric tensor power method of [19, Alg. 2] to $\hat{M}$ and invoking their perturbation bound (Theorem 5.1 therein) gives, for every component $r \in [4]$,

$$\big\|(\hat{a}_r, \hat{b}_r, \hat{c}_r) - (a_r, b_r, c_r)\big\|_2 = O\Big(\tfrac{1}{\delta} \|\hat{M} - M\|_{\mathrm{op}}\Big).$$

**Step 3: Assembling full means.** Algorithm 2 concatenates the three block-means, so $\hat{\mu}_r - \mu_r = (\hat{a}_r - a_r, \hat{b}_r - b_r, \hat{c}_r - c_r)$, and the same $O(\cdot)$ factor carries through.

**Step 4: Mixing-weight estimation.** Given accurate factor recovery, the usual least-squares re-estimation of weights satisfies $|\hat{\pi}_{qc} - \pi_{qc}| = O\big(\|\hat{M} - M\|_{\mathrm{op}}\big)$ (19, Theorem B.1), yielding the stated rate.

**Step 5: Union bound.** Combine Steps 1–4 and union-bound over the four components to obtain the final probability $1 - \varepsilon$. □

# E   Experiment Details

In this section, we provide experimental details and additional experiment results. We describe datasets details, evaluation prompts we used to collect LLM judgments, and individual judge performance. In addition, we introduce the construction of programmatic judge, and ablation studies including prompt-based interventions. Finally, we include additional experiment results for our tensor-based CARE algorithm: synthetic experiments results on graph-aware judge partition, and real-world applications.

## E.1   Datasets

**FeedbackQA [21].**   A question-answering dataset with human-provided scalar ratings of answer helpfulness, ranging from 1 to 5. We use the validation set in our experiments, treating the average of two human ratings as the ground truth.

**HelpSteer2 [23].**   A large-scale dataset of assistant responses annotated with real-valued scores (0 to 4) across multiple axes, including helpfulness, correctness, coherence, complexity, and verbosity. We use the validation set and take the helpfulness score as the ground truth.

**UltraFeedback [22].**   A scalar feedback dataset where assistant responses are rated from 0 to 10 based on overall quality, using scores aggregated from GPT-4 and human raters. We randomly sample 5,000 examples for evaluation.

**Synthetic Dataset (Section 5.6).**   We construct a synthetic dataset with latent state probabilities set to $\pi_{qc} = [0.2, 0.2, 0.3, 0.3]$, corresponding to latent states $(Q, C)$ as $(0,0), (0,1), (1,0), (1,1)$ respectively. The judges are organized into three distinct groups, each containing four judges whose conditional means $\mu_{qc}$ are randomly drawn from a uniform distribution ranging between 1 and 4. Dependence structures are imposed explicitly: for judges independent of the true quality variable $Q$, we constrain their conditional means such that averages depend solely on the confounder $C$ (i.e., rows corresponding to $Q = 0$ and $Q = 1$ are identical for each $C$ state).

## E.2 Prompt Templates

In this subsection we provide the prompts we used for collecting LLM judgements.

---

**LLM Judge Scoring Template (FeedbackQA, HelpSteer2, Ultrafeedback)**

You will be given a user_question and system_answer couple.

Your task is to provide a 'total rating' scoring how well the system_answer answers the user concerns expressed in the user_question. Give your answer as a float on a scale of {min_rating} to {max_rating}, where {min_rating} means that the system_answer is not helpful at all, and {max_rating} means that the answer completely and helpfully addresses the question.

Provide your feedback as follows:

Feedback:::
Total rating: (your rating, as a float between {min_rating} and {max_rating})

Now here are the question and answer.

Question: {question}
Answer: {answer}

Feedback:::
Total rating:

---

**Intervened LLM Judge Scoring Template (Section 5.4)**

You are a careful and impartial evaluator. Your task is to rate how well the system_answer responds to the user_question, focusing on substance, correctness, and completeness.

Rate the answer on a scale from {min_rating} to {max_rating}, where:
- {min_rating} means the answer does not address the question or is misleading,
- {max_rating} means the answer fully and helpfully addresses the question with accurate, relevant, and concise information.

Avoid common judgment biases:
- Do **not** reward irrelevant fluency, length, or politeness.
- Do **not** penalize answers solely for brevity if they are complete.
- Do **not** be influenced by phrasing style unless it affects clarity or correctness.

Judge only the answer's content in relation to the question. Ignore formatting, grammar, or tone unless they impact understanding or utility.

Provide your feedback as follows:

Feedback:::
Total rating: (a float from {min_rating} to {max_rating})

Now, here are the inputs:

Question: {question}
Answer: {answer}

---

**LLM Judge Toxicity Scoring Template (CivilComments)**

You will be given a user_question and system_answer couple.

Your task is to provide a 'total rating' scoring how well the system_answer answers the user concerns expressed in the user_question. Give your answer as a float on a scale of {min_rating} to {max_rating}, where {min_rating} means that the system_answer is not helpful at all, and {max_rating} means that the answer completely and helpfully addresses the question.

Provide your feedback as follows:

Feedback:::
Total rating: (your rating, as a float between {min_rating} and {max_rating})

Now here are the question and answer.

Question: {question}
Answer: {answer}

Feedback:::
Total rating:

## E.3 Individual LLM Judge Performance

Table 6 shows the performance of individual LLM judges. Overall, we find that aggregation methods outperform or match the best single judge. Since the relative performance of individual LLM judges is not known a priori in practical settings, aggregation offers a useful approach for integrating judges in an agnostic manner.

Table 6: Individual Judge Performance in Section 5.1

| | FeedbackQA | | HelpSteer2 | | UltraFeedback | |
|---|---|---|---|---|---|---|
| | MAE (↓) | $\tau$ (↑) | MAE (↓) | $\tau$ (↑) | MAE (↓) | $\tau$ (↑) |
| gemma-3-1b-it | 1.0073 | 0.2315 | 1.0666 | 0.0825 | 1.0606 | 0.1812 |
| gemma-3-4b-it | **0.7578** | 0.4537 | 0.9920 | 0.1402 | 0.8492 | 0.2309 |
| Llama-3.1-8B-Instruct | 0.8148 | 0.4341 | 1.1364 | 0.1261 | 0.8648 | 0.3194 |
| Llama-3.2-1B | 1.2219 | -0.0525 | 1.0049 | -0.0132 | 1.0119 | 0.0752 |
| Llama-3.2-3B | 1.0362 | 0.0051 | 0.9995 | 0.0251 | 1.1522 | 0.1648 |
| Mistral-7B-Instruct-v0.3 | 1.0244 | 0.4539 | 1.0793 | 0.1116 | 0.8572 | 0.1735 |
| Phi-4-mini-instruct | 0.8082 | **0.4557** | 1.0692 | 0.1576 | 0.8355 | 0.3147 |
| Qwen3-0.6B | 1.0969 | 0.2073 | 1.1255 | 0.0370 | 1.0233 | 0.1254 |
| Qwen3-1.7B | 1.1507 | 0.2485 | 1.0693 | 0.1049 | 1.1382 | 0.1926 |
| Qwen3-4B | 1.0999 | 0.2854 | **0.9675** | **0.2290** | **0.7088** | **0.3921** |
| Qwen3-8B | 1.0517 | 0.4417 | **0.9675** | 0.2094 | 0.7512 | 0.3140 |

## E.4 Programmatic Judges

Programmatic judges, synthesized by LLMs, distill and convert evaluation logic into interpretable, cheap-to-obtain program code [20, 61]. These program judges provide specialized, independent assessments compared to using LLMs directly as evaluators. We integrate these judge sets into CARE to enhance evaluation signals.

We describe the creation of programmatic judges and the criteria they encode. Using OpenAI's GPT-4o [33], we generate judges with the following prompt:

---

**Program Judge Template**

You are now a judge to evaluate LLM generated response with a given question. You will write your evaluation logic into code and generate python programs to return their scores. Higher represents better response quality. Consider complex criteria for assessing responses, leveraging third-party models, embedding models, or text score evaluators as needed.

Function signature: def _judging_function(self, question, response):

---

We synthesize 23 programs and select 10 representative ones for our experiments (see Section 5.2 and Section 5.3). These programs evaluate responses based on diverse criteria: (i) structure, (ii) readability, (iii) safety, (iv) relevance, and (v) factuality. For example:

- **Structure**: A judge counts transition markers (e.g., "therefore," "however") to assess coherence, with more markers indicating better quality.

- **Relevance**: A judge uses TF-IDF to convert questions and responses into vectors, computing cosine similarity to measure semantic alignment (see Program 1). Another employs semantic embeddings for similarity metrics (see Program 2).

- **Readability**: A judge leverages a third-party API to evaluate complexity, using metrics like the Flesch–Kincaid grade level (see Program 3).

All judging logic, conditions, and pre-defined keyword lists are generated by the LLM. Below, we provide examples to illustrate this approach.

```python
def _lexical_overlap(self, question, response):
    """Compute lexical overlap using TF-IDF for relevance evaluation.
        """
    # Preprocess input question and response (e.g., lowercase, remove
        stopwords)
    question_clean = self._preprocess(question)
    response_clean = self._preprocess(response)

    # Return 0.0 if either input is empty after preprocessing
    if not question_clean.strip() or not response_clean.strip():
        return 0.0

    # Transform inputs to TF-IDF vectors using the vectorizer
    tfidf_matrix = self.tfidf_vectorizer.fit_transform([question_clean
        , response_clean])
    question_vec = tfidf_matrix[0]   # Extract question vector
    response_vec = tfidf_matrix[1]   # Extract response vector

    # Compute cosine similarity between vectors and return as float
    return float(cosine_similarity(question_vec, response_vec)[0][0])
```

Program 1: Lexical Overlap Computation using TF-IDF.

```python
def _semantic_similarity_strong(self, question, response):
    """Compute semantic similarity between question and response."""
    # Return 0.0 if either input is empty
    if not question.strip() or not response.strip():
        return 0.0

    # Encode question and response into dense vectors using the
        embedding model
    question_embedding = self.semantic_embedding_strong_model.encode(
        question, max_length=4096
    )["dense_vecs"]
    response_embedding = self.semantic_embedding_strong_model.encode(
```

```
935          response , max_length =4096
936      )["dense_vecs"]
937
938      # Compute dot product similarity between embeddings
939      similarity = question_embedding @ response_embedding
940
941      # Clamp similarity score between 0.0 and 1.0 and return as float
942
943      return float(max(0.0, min(1.0, similarity)))
```

Program 2: Semantic Similarity using Embedding Model.

```
944
945  def _readability(self, response):
946      """Calculate readability metrics for response."""
947      # Compute readability scores using textstat library
948      return {
949          # Flesch Reading Ease (inverted: higher score means harder to
950          read)
951          "flesch_reading_ease": 100 - textstat.flesch_reading_ease(
952          response),
953          # SMOG Index (higher score indicates higher reading difficulty
954          )
955          "smog_index": textstat.smog_index(response),
956      }
957
```

Program 3: Readability Metrics Calculation.

We report the performance of individual program judges in Table 7. While their standalone performance is limited, they provide useful signals for the integration strategies discussed in Sections 5.2 and 5.3.

Table 7: Program Judge Performance. (*) represents the selected judges in Section 5.2.

| | FeedbackQA | | HelpSteer2 | | UltraFeedback | |
|---|---|---|---|---|---|---|
| | MAE ($\downarrow$) | $\tau$ ($\uparrow$) | MAE ($\downarrow$) | $\tau$ ($\uparrow$) | MAE ($\downarrow$) | $\tau$ ($\uparrow$) |
| factuality_check_score (*) | 1.9956 | 0.0872 | 1.1992 | 0.0075 | 1.1910 | 0.0492 |
| factuality_factKB_score (*) | 1.0343 | 0.2288 | 1.7180 | 0.0414 | 1.4342 | 0.1051 |
| readability_flesch_reading (*) | 1.2185 | 0.0431 | 2.5682 | 0.0445 | 2.5145 | 0.1396 |
| readability_smog (*) | 0.9805 | 0.1277 | 2.3286 | 0.0283 | 2.3122 | 0.1604 |
| relevance_bleu | 1.4035 | 0.0126 | 2.7452 | -0.0355 | 2.7330 | 0.0560 |
| relevance_keyword_overlap | 1.2779 | 0.1977 | 2.3735 | 0.0138 | 2.2725 | 0.1461 |
| relevance_lexical_overlap (*) | 1.1371 | 0.2316 | 2.0148 | -0.0144 | 1.9182 | 0.1187 |
| relevance_rouge | 1.3079 | 0.2066 | 2.5603 | 0.0232 | 2.4838 | 0.1327 |
| relevance_semantic_sim_strong (*) | **0.8759** | **0.4092** | 1.1182 | 0.0395 | **0.9866** | 0.1601 |
| safety_toxicity (*) | 1.5396 | -0.0380 | **1.1105** | 0.0300 | 1.0139 | -0.0043 |
| structure_avg_paragraph_length_dist | 1.4560 | -0.1883 | 2.5562 | -0.0081 | 2.4637 | 0.1074 |
| structure_avg_sentence_length_dist | 1.5248 | -0.0140 | 2.4407 | -0.0287 | 2.4179 | 0.1612 |
| structure_cohesion_score | 1.4078 | 0.2070 | 2.7139 | 0.0345 | 2.6578 | 0.1567 |
| structure_emphasis_count | 1.2826 | 0.1988 | 2.6642 | 0.0482 | 2.5955 | 0.2060 |
| structure_headings | 1.4765 | 0.0423 | 2.6521 | -0.0340 | 2.5916 | 0.1049 |
| structure_lexical_diversity | 1.0672 | 0.1625 | 2.1864 | 0.0444 | 2.0981 | 0.1935 |
| structure_list_usage | 1.6284 | 0.0159 | 3.0208 | -0.0108 | 3.0132 | 0.0872 |
| structure_logical_transitions (*) | 1.2694 | 0.1743 | 2.2693 | 0.0520 | 2.4355 | 0.2263 |
| structure_max_sentence_length (*) | 1.3039 | 0.1272 | 2.7532 | 0.0104 | 2.7511 | 0.1377 |
| structure_min_sentence_length | 1.3568 | 0.1887 | 2.4872 | 0.0400 | 2.4322 | 0.2046 |
| structure_questions | 1.2443 | 0.2692 | 2.4910 | 0.0360 | 2.4064 | 0.2114 |
| structure_sentence_balance | 1.4423 | 0.1835 | 2.6757 | 0.0501 | 2.6444 | 0.2203 |
| structure_sentence_count (*) | 1.3099 | 0.1742 | 2.4408 | **0.0807** | 2.6570 | **0.2300** |

## E.5  Effects of Prompt-Based Intervention (Section 5.4)

We begin by analyzing how the intervention using a robust prompt affects the performance of individual LLM judges. Figures 5 (MAE) and 6 (Kendall's $\tau$) present the performance differences relative to the vanilla prompt. While the intervention aims to reduce confounding signals, its impact varies—some model–dataset combinations show improvement, while others show degradation.

We then assess how these shifts influence aggregate performance. Figures 7 and 8 show the corresponding changes in aggregation accuracy. Most baseline methods benefit from the intervention, whereas CARE shows a slight performance drop. A plausible explanation is that once confounding signals are mitigated, the additional latent variables in CARE may begin to model residual noise rather than meaningful structure, slightly reducing its performance. Nevertheless, as shown in Section 5.4, CARE without intervention still outperforms other baselines with the robustness prompt, highlighting its effectiveness even without manually crafted interventions for hidden confounders.

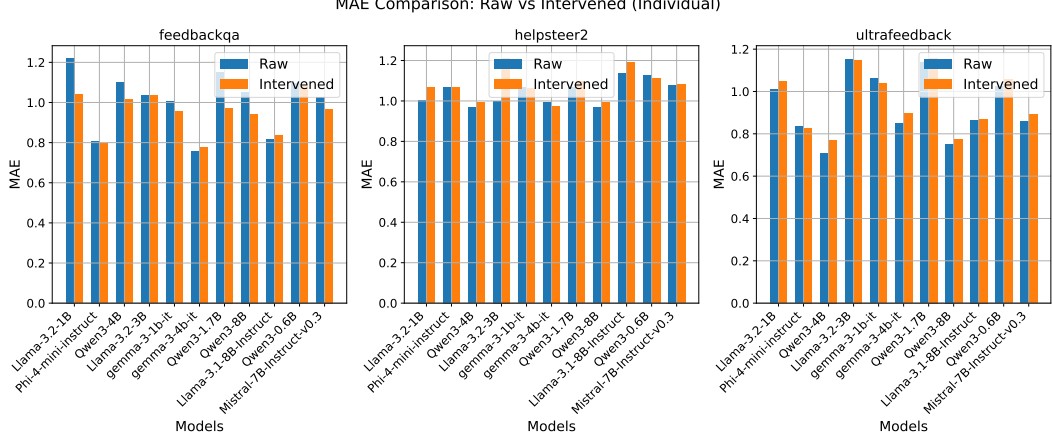

Figure 5: Change in MAE ($\downarrow$) for individual LLM judges after applying the robustness prompt.

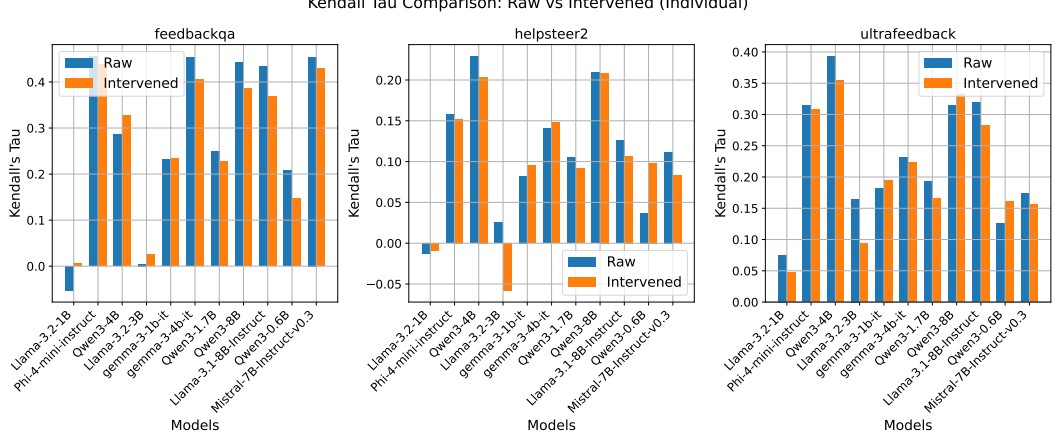

Figure 6: Change in Kendall's $\tau$ ($\uparrow$) for individual LLM judges after the robustness prompt.

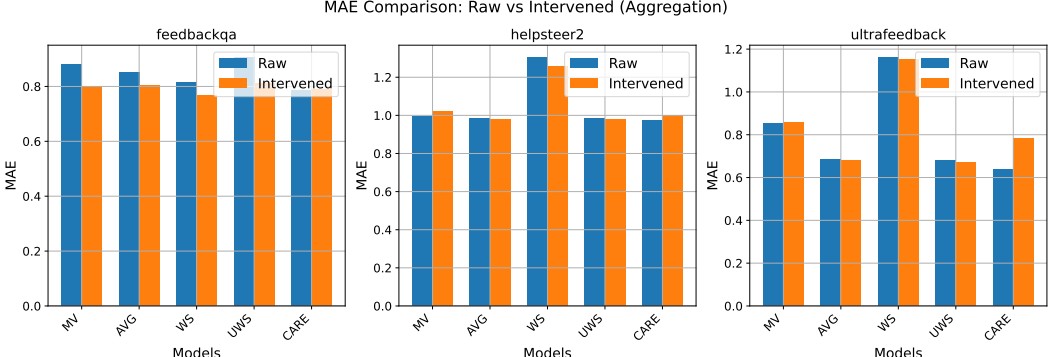

Figure 7: Change in aggregate MAE (↓) after propagating the robustness prompt through each aggregation method.

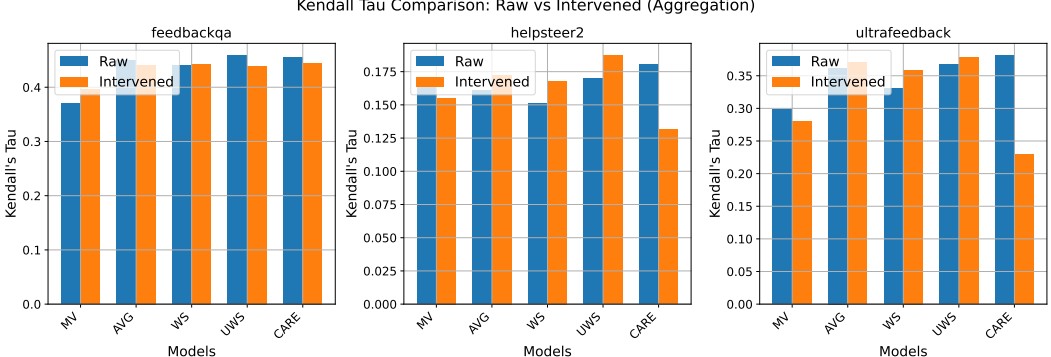

Figure 8: Change in aggregate Kendall's $\tau$ (↑) after the robustness prompt.

## E.6 Confounding Factors in Robustness Experiments

To clarify the setup in Section 5.5, we summarize the artificially injected confounding factors in Table 8, along with illustrative snippets. These perturbations target different dimensions of bias, ranging from superficial stylistic changes to alterations that directly affect semantic correctness.

| Bias | Perturbation Injected | Example Snippet (from Fig. 1) |
|---|---|---|
| *Fallacy Oversight* | Insert a factual error | "The square root of 36 is **7**..." (correct value is 6) |
| *Authority* | Add a fake citation | "...(Weisstein, Eric W. 'Square Root.' Math-World...)" |
| *Beauty* | Add emojis / formatting | "⑥ multiplied by ⑥ equals 36." |
| *Gender* | Add a gender-biased remark | "This might be a bit difficult for **women** to understand..." |

Table 8: Confounding factors and illustrative snippets.

Table 4 reports the robustness of different aggregation methods under these injected biases. We find that CARE is highly stable against stylistic biases such as *beauty* and *authority*, preserving both rankings and score magnitudes. In contrast, robustness deteriorates when the bias directly undermines factual or semantic content—as in *fallacy oversight* and *gender* perturbations.

This distinction aligns with our hypothesis: *fallacy oversight* introduces factual inaccuracies that reduce answer quality, producing expected shifts in judge scores. Meanwhile, *gender* bias activates explicit safety mechanisms in alignment-tuned LLM judges, leading to consistent downscoring across models and correspondingly large shifts in aggregate outcomes.

## E.7 Additional Controlled Experiment on Confounding Factors

Unlike the semi-synthetic perturbations in Section 5.5, here we investigate whether CARE can separate the true quality latent factor from naturally arising confounders in a more controlled setting. Specifically, we introduce two dummy judges whose scores are directly correlated with response length or the presence of specific words. If CARE functions as intended, CARE should recover a factor structure in which high-quality judges align with the true quality factor $Q$, while the dummy judges align with a distinct confounder.

**Setup.** We ran CARE-SVD with 14 judges on the FEEDBACKQA dataset, combining 10 LLM judges, 2 programmatic "dummy" judges (sensitive to length or special keywords), and 2 human annotators. The factor loadings are presented in Table 9.

**Results.** The observed loadings align with our hypothesis:

- **Factor 1 (true quality $Q$).** This factor exhibits *broad, balanced loadings* across competent LLM judges and the two human judges, with much weaker loadings for the programmatic dummy judges. Within model families, larger models have higher loadings (e.g., Llama-3.1-8B > Llama-3.2-3B ≈ Llama-3.2-1B), suggesting that $Q$ reflects underlying capability. Instruction-tuned models (Mistral-7B-Instruct, Phi-4-mini-instruct, Llama-3.1-8B-Instruct, Gemma-3-4B-it) also show above-median loadings, consistent with their alignment to human rubrics.
- **Factor 2 (length confounder).** This factor is dominated by a *high, concentrated loading* on the length-sensitive dummy_eval_1, with a secondary loading on gemma-3-1b-it (0.59). In contrast, nearly all other judges—including both humans and stronger instruction-tuned models—have near-zero loadings. Such a one-sided, few-judge pattern is characteristic of a confounder rather than true quality.

Table 9: Judge loadings on latent factors in CARE-SVD. Factor 1 corresponds to true quality $Q$; Factor 2 reflects a length confounder.

| Judge | $Q$ (true quality) | Length confounder |
|---|---|---|
| Qwen3-8B | 0.396 | -0.240 |
| Llama-3.1-8B-Instruct | 0.664 | -0.076 |
| gemma-3-4b-it | 0.706 | -0.152 |
| Llama-3.2-1B | -0.009 | -0.140 |
| Qwen3-4B | 0.180 | 0.008 |
| gemma-3-1b-it | 0.243 | 0.595 |
| Llama-3.2-3B | 0.033 | 0.057 |
| Phi-4-mini-instruct | 0.715 | -0.051 |
| Qwen3-1.7B | 0.199 | -0.012 |
| Mistral-7B-Instruct-v0.3 | 0.804 | 0.016 |
| dummy_eval_1 | 0.098 | 0.742 |
| dummy_eval_2 | 0.035 | 0.290 |
| human_eval_1 | 0.337 | 0.078 |
| human_eval_2 | 0.338 | 0.059 |

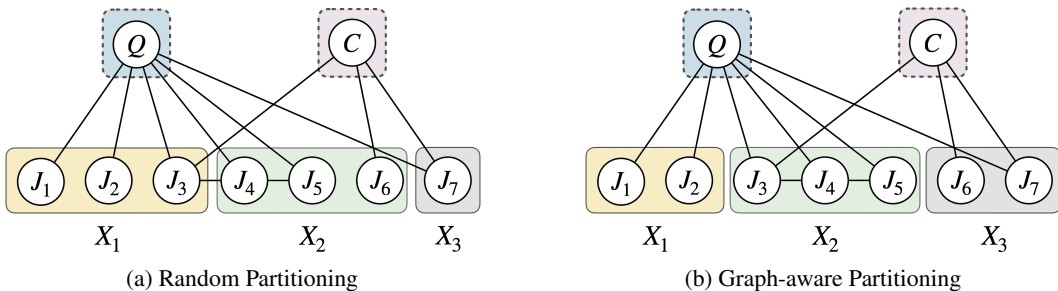

| (a) Random Partitioning | (b) Graph-aware Partitioning |

Figure 9: Random Partitioning vs. Graph Aware Partitioning. A random partitioning (a) leaves cross-view edges that violate the independence assumptions of tensor methods, whereas the graph-aware partitioning (b) considers cross-view edges and restores the required separation.

## E.8 Additional Real-World Experiment on Gaussian Mixture

We consider a Gaussian mixture setting where the latent variable is binary, but the observables (judge outputs) are real-valued Gaussian scores. This experiment evaluates the effectiveness of Algorithm 2 on a real dataset.

**Setup.** We use a subset of the CivilComments dataset [62], randomly sample 5,000 examples. The ground-truth label is binary toxicity (0 or 1), while LLM judges provide real-valued toxicity scores ranging from 0 to 9. In addition to the original LLM judges, we include five LLMs:

- `meta-llama/Meta-Llama-3-8B-Instruct`,

- `mistralai/Mistral-7B-Instruct-v0.2`,

- `Qwen/Qwen2.5-0.5B-Instruct`,

- `Qwen/Qwen2.5-1.5B-Instruct`,

- `Qwen/Qwen2.5-3B-Instruct`.

For the MV and WS baselines, we first discretize judge scores using a threshold of 4.5 before applying majority vote or weighted sum. For AVG and UWS, we aggregate scores first, then apply the threshold. CARE (Algorithm 2) directly infers the latent binary label from continuous scores. We evaluate all methods using classification accuracy.

Table 10: Aggregated accuracy (higher is better) in CivilComments dataset.

| Method | Acc. (%) |
|--------|----------|
| MV     | 74.32%   |
| AVG    | 73.80%   |
| WS     | 74.95%   |
| UWS    | 74.95%   |
| CARE   | **75.27%** |

**Results.** Table 10 shows that CARE achieves the highest accuracy. This result highlights its ability to better handle confounding factors and perform effective latent inference, even when the observed data (continuous scores) differ from the latent variable type (binary labels).

## E.9 Synthetic Experiment on Graph-Aware Tensor Decomposition

When judges exhibit conditional dependencies, naively partitioning them into views violates the independence assumptions required by tensor decomposition. We hypothesize that partitioning judges via a graph-aware procedure that respects dependency structure yields substantially better estimation than random partitioning.

**Setup.** We simulated 10,000 items scored by $p = 12$ judges, partitioned into three views of four judges each. To induce conditional dependencies, we planted edges of strength 0.3 within each true view at 40% density. We then compared two grouping strategies across ten random seeds:

- **Random**: assign judges to views uniformly at random;

- **Graph-Aware**: assign views to minimize cross-block edges in the empirical precision matrix.

Performance was measured by the $\ell_2$ error in recovering the latent component means, i.e. $||\mu_{qc} - \hat{\mu}_{qc}||_2$).

**Results.** As shown in Figure 10, graph-aware grouping dramatically reduces reconstruction error—by more than an order of magnitude—compared to random grouping. This confirms the importance of respecting dependency structure during view formation and underscores the advantage of CARE, which integrates graph structure directly into the tensor decomposition procedure.

### E.10 Computing Resources

We used a server equipped with an NVIDIA RTX 4090 (24GB). Generating LLM judge outputs took up to 3 hours per dataset. In contrast, the aggregation algorithms were efficient, completing in under 1 minute for datasets with approximately 5,000 rows.

Figure 10: $\ell_2$ reconstruction error (mean $\pm$ SD) for random vs. graph-aware grouping.

## F   Broader Impact Statement

This work presents a novel approach to aggregate scores from multiple LLMs serving as judges by identifying confounding variables and thus potentially reducing the bias in the overall judge scores. The potential broader impact includes a framework for improved LLM-as-a-judge scores which can be used at various applications. However, it is important to acknowledge that using LLMs as potential judges to automate labor-intense annotation tasks which is an active area of research carries some limitations and past research has discussed some unintended consequences, such as over-reliance on judge outputs, misuse and misinterpretation of results which might carry high real-world stakes. It is crucial to use automated LLM-as-a-judge tools responsibly and ethically, considering potential biases in data and models, and ensuring transparency and accountability in their application.

