# OpenReview forum: "From Many Voices to One: A Statistically Principled Aggregation of LLM Judges"
_NeurIPS.cc/2025/Workshop/Reliable_ML — NeurIPS 2025 - Reliable ML Workshop_

### Official Review · Reviewer_AJgh · 2025-09-09
**timely topic!**

**Rating:** 7
**Confidence:** 2

**Review:**

[Summary]

This paper suggests a new method to aggregate multiple LLM-as-a-judge. The authors assume a latent variable structure with markov random fields, and try to estimate the latent factors using sparse+low-rank decomposition of the score matrix. Then, they eventually obtain the true quality factor by taking a Bayesian philosophy with heuristics. This approach is tested on multiple datasets and demonstrates its strength in many tasks.

[Strengths] Novelty, rigor, empirical/theoretical quality, clarity, relevance to reliability with imperfect data.

1. Hot topic!
2. Empirically validated in many datasets.
3. The main algorithm is clearly defined.

[Weaknesses / Limitations]

1. The current writing is not really delivering the research very well. The abstract doesn't flow well, and is unnecessarily wordy. Try to deliver high-level idea in the abstract, not too much details in there. Also, the paper never defines what LLM-as-a-judge is until the end of the second page. While it is obvious to the authors, it would be nice to clearly define what it is in the beginning. Likewise, it will help the readers if the paper is self-contained by introducing all nontrivial concepts. Lastly, related work showing up in the last page is weird to me (but perhaps it is more common in llm papers?), I would want to see this in intro....
2. I like low-rank + sparse approximation, but why is it appropriate in this particular use case? How does it overcome C1 and C2? (I think page 4 tries to address this but i am not convinced yet.)

[Suggestions]
1. Define f in Algorithm 1, line 1.
2. It would be nice if the paper explicitly explained the experimental setting (e.g., what is in the dataset and how big it is, etc. I am assuming texts with preference labels? or is it different?) and how the scores (MAE and tau) are calculated.

---

### Official Review · Reviewer_y3A5 · 2025-09-19
**Well-written, clear paper attempting to solve a real problem, perhaps too appendix-happy and would be stronger with a discussion on transitioning to practice**

**Rating:** 7
**Confidence:** 1

**Review:**

## Summary

The authors attempt to mitigate the biases, correlations, confounders that can affect multi-model judge systems. They present the CARE framework that models the (not obvious) confounding factors for each judge. The framework is able to reliably aggregate judge scores while accounting for the aforementioned weaknesses.

## Strengths and Weaknesses

* Well written, clear
* The underlying problem is significant
* The proposed solution, while it synthesizes existing concepts, is sufficiently novel
* Appendix B, parts of C, and parts of E likely belong in the main body of the paper

## Suggestions

* Consider discussing more of what's in the appendix in the main body of the paper
* Add a discussion about practical applications; i.e., where do we typically see LLM ensembles acting as judges where CARE would be valuable, and any experiences making that transition

## Ethics

Fundamentally, this paper considers the question of how to get "true" assessment from LLMs. This is highly subjective and may have ethical implications. The authors do call this out in appendix F.